# Crystal structure of the Na⁺/H⁺ antiporter NhaA at active pH reveals the mechanistic basis for pH sensing

Iven Winkelmann[1,4], Povilas Uzdavinys[1,4], Ian M. Kenney [2,4], Joseph Brock [1], Pascal F. Meier[1], Lina-Marie Wagner[1], Florian Gabriel[1], Sukkyeong Jung [1], Rei Matsuoka[1], Christoph von Ballmoos[3], Oliver Beckstein [2] ✉ & David Drew [1] ✉

The strict exchange of protons for sodium ions across cell membranes by Na⁺/H⁺ exchangers is a fundamental mechanism for cell homeostasis. At active pH, Na⁺/H⁺ exchange can be modelled as competition between H⁺ and Na⁺ to an ion-binding site, harbouring either one or two aspartic-acid residues. Nevertheless, extensive analysis on the model Na⁺/H⁺ antiporter NhaA from *Escherichia coli*, has shown that residues on the cytoplasmic surface, termed the pH sensor, shifts the pH at which NhaA becomes active. It was unclear how to incorporate the pH senor model into an alternating-access mechanism based on the NhaA structure at inactive pH 4. Here, we report the crystal structure of NhaA at active pH 6.5, and to an improved resolution of 2.2 Å. We show that at pH 6.5, residues in the pH sensor rearrange to form new salt-bridge interactions involving key histidine residues that widen the inward-facing cavity. What we now refer to as a pH gate, triggers a conformational change that enables water and Na⁺ to access the ion-binding site, as supported by molecular dynamics (MD) simulations. Our work highlights a unique, channel-like switch prior to substrate translocation in a secondary-active transporter.

The strict exchange of Na⁺ for H⁺ ions is a fundamental process that was first described by Peter Mitchell and colleagues in the 1970's[1]. Na⁺/H⁺ antiporters are found in all cells to help regulate intracellular pH, sodium levels, and cell volume[2,3]. They are associated with a number of different diseases such as neurological disorders, diabetes, and cancer[2,4,5] and they are of growing importance as drug targets[2,6]. To study the molecular basis of their function, bacterial homologues are often used as they are easier to manipulate. In bacteria, they use the proton-motive force to extrude sodium out of the cell contributing to cell homeostasis and the generation of a

sodium-motive force, making them also an important class of protein for pathogenic bacteria[7].

Bacterial NhaA is an electrogenic Na⁺/H⁺ antiporter that transports 2 H⁺ in exchange for 1 Na⁺ across the membrane[8,9]. *E. coli* NhaA (*Ec*NhaA) was one of the first secondary-active transporters to be characterised at the structural level[10], initially by electron 2D-crystallography[10,11] and later by X-ray crystallography[12], and together with extensive biochemical analysis has established itself as the major model system for establishing mechanisms for sodium/proton exchange[3]. The NhaA-fold, as first described in *Ec*NhaA, and later seen

[1]Department of Biochemistry and Biophysics, Science for Life Laboratory, Stockholm University, SE-106 91 Stockholm, Sweden. [2]Center for Biological Physics and Department of Physics, Arizona State University, Tempe, AZ 85287, USA. [3]Department of Chemistry and Biochemistry, University of Bern, Freiestrasse 3, 3012 Bern, Switzerland. [4]These authors contributed equally: Iven Winkelmann, Povilas Uzdavinys, Ian M. Kenney. ✉e-mail: obeckste@asu.edu; ddrew@dbb.su.se

in other bacterial Na$^+$/H$^+$ antiporters[3,10,12] has also been observed in unrelated bile acid sodium-coupled symporters and bicarbonate transporters[13,14]. The NhaA-fold consists of a scaffold domain that mediates dimerization (dimer domain), and an ion-translocation (core) domain, which is typified by a six transmembrane (TM) bundle harbouring two opposite-facing discontinuous helices that crossover near the centre of the membrane[3,12].

The first EcNhaA crystal structures was determined as a monomer (referred to as mono-NhaA) at 3.45 Å at pH 4[12] and then later as a physiological homodimer[15] (referred to as dimer-NhaA)[16] at 3.5 Å resolution at pH 4. Further structures of the electrogenic Na$^+$/H$^+$ antiporter NapA from Thermus thermophilus (TtNapA) were later determined in both inward- and outward-facing conformations at active pH, showing that alternating access to the ion-binding site was achieved by elevator-like transitions of the core domain[17,18], rather than local changes in half-helices[12,19,20]. Although the outward-facing crystal structure of TtNapA was determined at a higher resolution of 2.3 Å and at active pH, there was no evidence of Na$^+$ bound in the structure[18]. Despite extensive biochemical, biophysical, structural and computational investigation, the question how proton and sodium transport are strictly coupled remains an active area of debate. Key to answering this question is to establish the identities of specific residues responsible for controlling ion translocation in the transport cycle.

EcNhaA is inactive below pH 6.0, and between pH 6.5 and pH 8 transport rates increase 2000-fold[9]. The transition from inactive to active pH is associated with increased dynamics that match pH-dependent transport as demonstrated by trypsin digestion profiles[21,22], cysteine-accessibility measurements[23,24], antibodies specific to only the state at active pH[25,26], tryptophan fluorescence[27], single-molecule force spectroscopy profiles[28,29], and structural rearrangements observed by 2D-electron crystallography[19]. A proposed pH sensor model based on the crystal structure at low pH[12], computational analysis[30] and pH activity measurements of variants[23,24,31–33], has been mapped to a charged network located at the cytoplasmic surface. However, despite exhaustive experimentation, the mechanistic basis for pH sensing in EcNhaA remains unclear, and difficult to combine with an alternating-access mechanism that at active pH fits a kinetic model of H$^+$ vs. Na$^+$ competition to the ion-binding site[34].

Here we determine the crystal structure of EcNhaA at active pH 6.5 at 2.2 Å resolution. We show that between inactive and active pH, EcNhaA undergoes a channel-like opening of the intracellular funnel to enable Na$^+$ to access the strictly-conserved ion-binding aspartate Asp164. We further show that the neighbouring Asp163 forms a salt-bridge to Lys300 at both inactive and active pH, but only at active pH do molecular dynamics (MD) simulations show that Na$^+$ binding induces salt-bridge breakage, an essential step in the transport cycle.

## Results

### Lipidic cubic phase (LCP) crystal structure of NhaAWTlike triple mutant at 2.2 Å resolution

We previously reported the structure of dimeric EcNhaA for a triple-mutant (Ala109Thr, Gln277Gly, Leu296Met) with greater thermostability than wildtype EcNhaA[16]; the crystal structure of the EcNhaA triple mutant and the pH dependence profile and kinetics were indistinguishable from wildtype (WT)[16]. We further uncovered that EcNhaA was stabilized by the lipid cardiolipin, which could be used to compensate for the destabilization effect caused by monoolein during lipidic-cubic phase (LCP) crystallization[35]. LCP crystals of the EcNhaA WT-like mutant were grown at active pH 6.5 and the structure phased by molecular replacement and refined to 2.2 Å resolution, R$_{factor}$ 20.9% and R$_{free}$ 23.5% (Fig. 1a and Table 1) (PDB ID: 7S24; here referred to as monoLCP-NhaA). Notably, although the NhaA WT and the NhaAWT-like triple mutant is only ~10% active at pH 6.5, as assessed by proteoliposome-based assays upon the addition of 10 mM LiCl[16], pH-induced changes as detected by 2D-electron crystallography[19] and

atomic force measurements[28] are already apparent for EcNhaA WT from pH 6.0 in detergent. The monoLCP-NhaA crystallized as a non-physiological monomer, like the previous mono-NhaA crystal structure determined at 3.45 Å resolution at inactive pH 4[12]. Overall, there are small, but clear differences between EcNhaA crystal structures determined under inactive pH 4 and active pH 6.5 conditions, which are spread across several helices in both the dimer and core domains (Fig. 1b and Supplementary Fig. 1a, b).

In addition to structural differences, non-protein lipid-like densities were identified in the interface between the core and dimer domains, in a position similar to observed crystallization lipids in the TtNapA LCP crystal structure (Supplementary Fig. 2a). Previously, lipid-like density was also observed between protomers in the dimeric-NhaA structure that was thought to represent the binding site for cardiolipin[35–37]. Consistent to cardiolipin preferring to bind to the dimeric form of the protein, no density to support any tightly-bound lipid was seen in this proposed location in the monoLCP-NhaA structure, despite its addition during purification (Supplementary Fig. 2b). Mutagenesis of the suspected cardiolipin binding residues Arg203 and Arg204 at the dimerization interface both to alanine, nonetheless, abolished binding as assessed by GFP-TS, which was used previously to monitor cardiolipin-specific binding to EcNhaA (Supplementary Fig. 2c)[35]. Indeed, the loss of cardiolipin binding upon mutagenesis of Arg203 and Arg204 to alanine is consistent with in vivo data that showed these residues were the most important for NhaA homo-dimerization and function[38]. It seems whilst cardiolipin increases stability of the dimer and aided LCP crystallization, EcNhaA still preferred to crystallize as a monomer under these non-physiological conditions. In addition, 106 crystallographic waters were modelled with one of the waters in hydrogen bond distance to the ion-binding Asp163 (Supplementary Fig. 2a).

### Conformational changes in the pH gate causes widening of the intracellular funnel

The pH sensor in NhaA is a charged network of residues clustered towards the cytoplasmic funnel (Fig. 1a). Of importance, the mutation of TM9 residues Glu241, Gly242, Arg250, Glu252, His253, His256, Val254 and TM2 residues Asn64, Asp65, Leu67, Glu78, Glu82 to cysteine shifts the pH at which NhaA becomes active (Supplementary Table 1)[22–24,33]. Consistent with biochemical analysis[39], conformational changes upon activation are centred around TM9, the TM8-TM9 loop, and the cytoplasmic ends of TM2 and TM1 (Fig. 1c).

In the NhaA crystal structure at active pH 6.5, Arg81 in TM2 breaks its salt-bridge interaction with Glu252 in TM9 and forms a tighter salt-bridge (>3.3 Å) to Glu78 located on its own helix, stabilized by Glu82 and Glu252 through interactions with Glu78 (Fig. 1c and Supplementary Fig. 1b). We further observe a rearrangement of His253 and His256 in TM9. In particular, the rearrangement of Asp11 in TM1 has broken its interaction with His256 in TM9 and reformed a salt-bridge to Lys153 in TM5, which has also shifted its position. Since the ion-binding Asp164 is located on TM5, there is therefore a direct physical coupling between the pH induced changes and the ion-binding site (Fig. 1b, c). The newly formed Asp11-Lys153 salt-bridge is consistent with the conformational stabilization of TM1 observed by 2D electron crystallography[19] and epitope mapping of antibodies at active pH[26]. The Asp11-Lys153 salt-bridge displaces the position of the TM4 harbouring Phe136, which is also an important indicator of pH activation, as monitored by tryptophan fluorescence (Supplementary Fig. 2d)[27]. Lastly, the movement of Lys249 and Glu252 in TM9 are consistent with the trypsin-sensitive cleavage of Lys249 at alkaline pH[22] and the Glu252Cys mutant, for which its accessibility to thiol probes follows the activation pH profile of NhaA[23]. The concerted movement of Arg81, His256 and Asp11 side-chains at active pH has caused a dramatic widening of the intracellular funnel (Fig. 1d and Supplementary Movie 1–3). The ion-

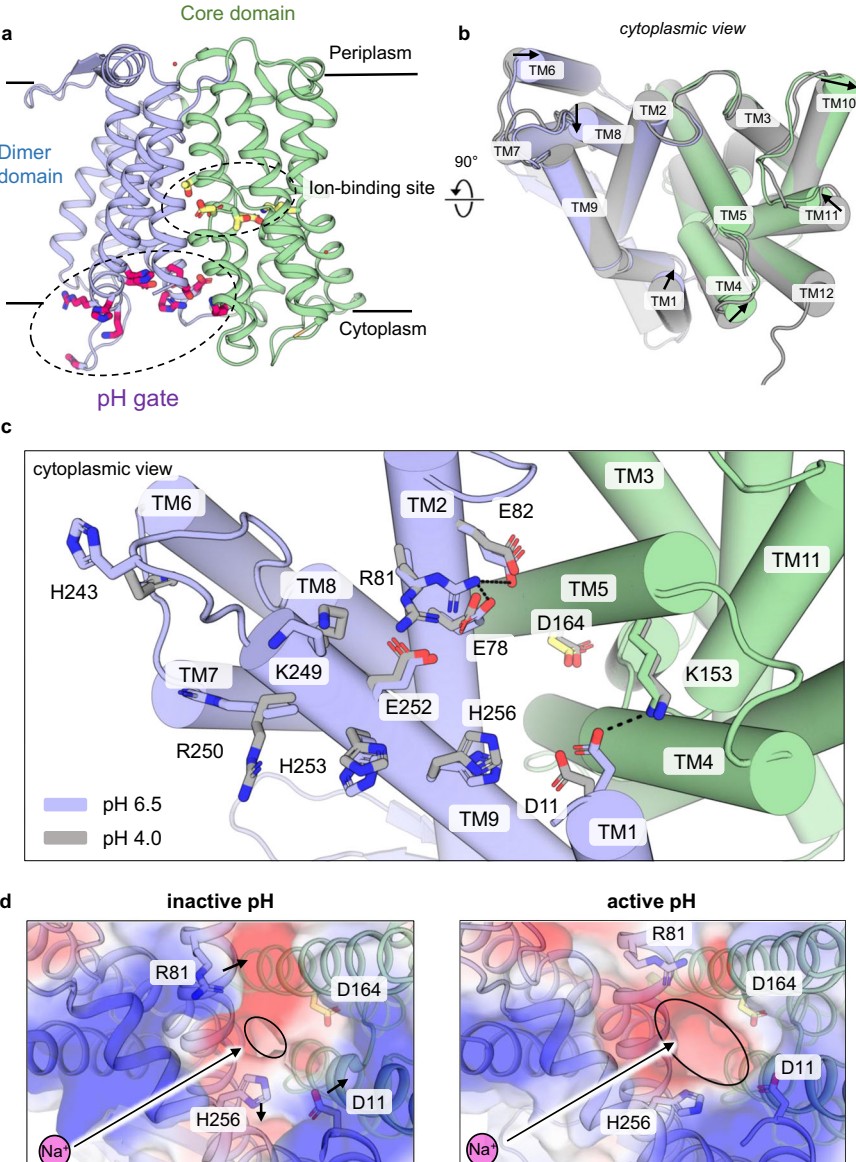

**Fig. 1 | Crystal structure of NhaA at active pH 6.5 reveals the pH sensor is a switch to open access to ion-binding site. a** Cartoon representation of monoLCP-NhaA. Showing the dimer domain (blue), 6TM core domain (green), pH sensing region residue (as pink sticks), and ion-binding site residues (as yellow sticks). **b** Cylindrical cartoon representation of NhaA from the cytoplasmic site, overlaying the pH 4.0 structure (PDB ID: 4AU5, grey) over the pH 6.5 structure (PDB ID: 7S24, colored). Movement of helices between the two structures are indicated by arrows. **c** Look onto the cytoplasmic pH-sensing region of *Ec*NhaA. Residues identified to

play a crucial role in pH-sensing are shown as sticks and salt-bridges indicated by a dashed line. The pH 4.0 structure (PDB ID: 4AU5, grey) is superimposed against the monoLCP-NhaA structure (coloured). **d** Electrostatic surface representation looking at the inside-open ion-binding funnel of *Ec*NhaA, comparing the dimerLCP-NhaA structure (PDB ID: 4AU5) at inactive pH 4.0 (left) against the monoLCP-NhaA structure at pH 6.5 (right). Arrows indicate the rearrangements of indicated amino acids. The ion-binding funnel is marked by a black ellipsis. It notably widens at active pH.

binding Asp164 in TM5 at active pH has further moved 1 Å towards the inward-facing funnel (Supplementary Fig. 2e).

Based on structural analysis and MD simulations it was previously concluded that the funnel of NhaA at pH 4 was too narrow to enable the passage of a hydrated $Na^+$ beyond the Asp11 and Glu252 residues[12,40]. Consistently, we find that only at active pH 6.5 is the intracellular funnel in NhaA wide enough to accommodate a hydrated $Na^+$ ion to the ion-binding site, as assessed by HOLE analysis (Fig. 2a, b). To complement the analysis of static structures, MD simulations were carried out starting from the high-resolution structure determined at active pH (see Methods). At active pH 7.5, the cytoplasmic funnel remained open and allowed both $Na^+$ ions and water to enter the binding site (Fig. 2c, d), even if the ion-binding site was modelled with two protons bound (corresponding to a local pH 4 in the binding site)

and cannot bind $Na^+$. In contrast, under inactive pH conditions, the cytosolic funnel contracted over the course of the simulation with concomitant changes in salt bridge interactions (Fig. 2c), preventing $Na^+$ access to the binding site (Fig. 2d). Taken together, we conclude the pH sensor is rather working as a pH gate, controlling access to the ion-binding site.

## Mechanistic basis for the pH gate centres around histidine residues

Although the pH gate region has been well-documented and studied, Asp11 and Lys153 residues were yet to be assessed by mutagenesis and whilst His253 and His256 have predicted $pK_a$ values to allow protonation/deprotonation events in the range between pH 5−7.5[40], these residues have only been assessed by replacement with

**Table 1 | Data collection, phasing and refinement statistics**

| | Mono-LCP NhaA |
|---|---|
| *Data collection* | |
| Space group | *P1* |
| *Cell dimensions* | |
| a, b, c (Å) | 46.5, 47.3, 56.4 |
| α, β, γ (°) | 78.0, 67.1, 79.2 |
| Resolution range (Å) | 34.84–2.2 (2.3–2.2) |
| Total reflections | 74862 (7733) |
| Unique reflections | 19902 (2006) |
| $R_{-merge}$ (%) | 10.18 (105.5) |
| $R_{-pim}$ | 0.062 (0.626) |
| $CC_{1/2}$ | 1.00 (0.53) |
| CC* | 1.00 (0.83) |
| I / σI | 7.22 (0.90) |
| Multiplicity | 3.8 (3.8) |
| Completeness (%) | 91.6 (92.9) |
| *Refinement* | |
| $R_{work}/R_{free}$ (%) | 20.9/23.5 |
| *No. atoms* | |
| All (non-hydrogen) | 2923 |
| Protein | 2799 |
| Others (PEG, lipid) | 66 |
| Solvent | 58 |
| *B-factors ($Å^2$)* | |
| All | 57.9 |
| Protein | 57.4 |
| Others (PEG, lipid) | 78.2 |
| Solvent | 55.4 |
| *R.m.s deviations* | |
| Bond lengths (Å) | 0.004 |
| Bond angles (°) | 0.64 |

Statistics for the highest-resolution shell are in parentheses.

cysteine[24] (Supplementary Table 1). The residue His256 is of particular interest as it connects the charged surface residues with core domain residues (Figs. 1c and 3a). Indeed, in a computational study, His256 was noted as the main residue strongly connecting the extracellular pH sensor with core domain residues[41]. To gain further insights into the interaction network, we performed all-atom MD simulations with the active pH structure. We varied the protonation states to model either the low pH (~3.5, inactive) or high pH (~7.5, active) with functionally relevant protonation states in the binding site (see Methods for details). Either sets of simulations display little conformational drift with low $C_\alpha$ RMSDs below ~2.5 Å, relative to the crystal structure and well-maintained secondary structure (Supplementary Fig. 3 and Supplementary Fig. 4). In the low pH MD simulations, Asp11 was found to form a salt bridge to His256 (Fig. 3a, b), which is likely because His256 is positively-charged at low pH with a predicted $pK_a$ of 6.9[40]. The neighbouring His253, with a predicted $pK_a$ of 6.3, favoured salt-bridge formation to Glu252 (Supplementary Fig. 6a), whereas Arg81 frequently interacted with either Glu78/Glu82 or Glu252 (Supplementary Fig. 5). Contrastingly, at high pH, Arg81 fluctuated between interacting with either Glu78 or Glu252, as Glu252 was no longer interacting with the neutral His253 at active pH (Supplementary Fig. 6). On the other side of the funnel, Asp11 switched to form a salt-bridge to Lys153, and no longer interacted with the now neutral His256 (Fig. 3b). Overall, the MD simulations support the rearrangements observed at active pH, and

the judicial placement of histidine residues linking both sides of the cytoplasmic funnel.

Solid-state membrane (SSM) electrophysiology measurements of *Ec*NhaA display transient currents down to pH 5.0 in the presence of an outwardly-directed proton gradient[34] i.e., the cavity in NhaA must be wide enough at pH 5.0 to enable Na⁺ translocation, even if NhaA cannot effectively catalyse proton efflux at pH 5.0. This interpretation is consistent with recent[22] [Na⁺] binding assays, that showed half-maximal binding of Na⁺ to *Ec*NhaA already at pH 6.0[42]. Indeed, in proteoliposome assays acidified by the ATPase, the triple-mutant *Ec*NhaA has clear activity at pH 6.5 upon the addition of external 150 mM LiCl, as assessed by dequenching of the pH-sensitive dye 9-Amino-6-Chloro-2-Methoxyacridine (ACMA) (Fig. 3c). To assess the impact of Asp11, Lys153, His253, His256 and Glu78 residues more thoroughly, alanine substitutions were generated and purified, showing similar yields and quality as the NhaA-WT-like triple mutant (Supplementary Fig. 7a, b). In proteoliposome-based assays, the acidic residue substitutions Asp11Ala and Glu78Ala had different pH activation profiles (Fig. 3d). Compared to the NhaA WT-like triple mutant, the Glu78Ala variant demonstrated a stronger response to LiCl addition at pH 7.0, with maximal activity at pH 7.5, rather than the maximal activity at pH 8.5 for wildtype[43]. In contrast, the Asp11Ala variant had poor activity at pH 7.5 with activity only increasing at pH 8.5. At pH 7.5, the apparent $K_M$ of Glu78Ala and Asp11Ala variants were similar to the NhaA-WT-like triple mutant (Fig. 3e). The variants alter the pH activation profile, even if they do not themselves influence ion-binding as inferred from their Michaelis–Menten affinities (Fig. 3e). The variants His253Ala, His256Ala, Lys153Ala and Lys153Gln had a drastic effect on NhaA activation (Fig. 3f, Supplementary Fig. 7c). His253Ala and Lys153Ala had almost no measurable activity until pH 8.5 and furthermore the activity was very low with ~10% dequenching. The His256Ala and Lys153Gln variants showed very poor activity altogether. The His256Ala and Lys153Gln activities were too low to accurately estimate an $K_M$, whereas His253Ala and Lys153Ala variants have an apparent $K_M$ 1.5 to 2-fold higher than NhaA wildtype[16]. Taken together, the protonatable histidine and lysine residues in the pH gate are of critical importance for NhaA activation in the proteoliposome assays, most likely as their charged states are required to stabilise an open inward-facing cavity.

### The Asp164-Lys300 Na⁺ sensitive salt-bridge

Based on sequence conservation, mutational analysis and the mono-NhaA crystal structure, the conserved residues Asp163 and Asp164 on TM5 in the 6TM-core domain were modelled to be the main ion-binding site residues and proton carriers conveying electrogenicity to the transporter[3,12,20,44]. In the dimer-NhaA crystal structure, however, TM10 was remodelled, resulting in a re-positioned Lys300 that formed a salt-bridge with Asp163[16]. Molecular dynamics (MD) simulations based on the remodelled NhaA crystal structure suggested Lys300, and not Asp163, as the second proton carrier[16,40]. This proposal was consistent with biochemical data of *Tt*NapA[43] and analysis of constant-pH MD simulations of the dimer-NhaA structure[40,45], leading to the proposition of a new transport model. In this model, one H⁺ is bound to Asp164 and the second H⁺ is bound to the protonated Lys300, which is salt-bridged to Asp163 in the outward-facing state. Once inward-facing, Na⁺ binding is thought to break the Asp163-Lys300 salt-bridge.

In the high-resolution structure of the NhaA WT-like triple mutant at active pH 6.5, Lys300 is forming a salt-bridge to Asp163 (Fig. 4a), as was previously modelled in the dimeric-NhaA structure at low pH[16]. In the monoLCP-NhaA, there is also a well-resolved water nearby, which would be well positioned to enable the hydration of Lys300 upon salt-bridge breakage (Fig. 4a). To assess the stability of the Asp163-Lys300 salt bridge of the high-resolution structure, we carried out all-atom MD simulations of NhaA for multiple protonation states of Asp163 and Asp164. In MD simulations, Na⁺ binds only when Asp164 and Asp163 are deprotonated as seen by ion-Asp carboxylate distances <3 Å

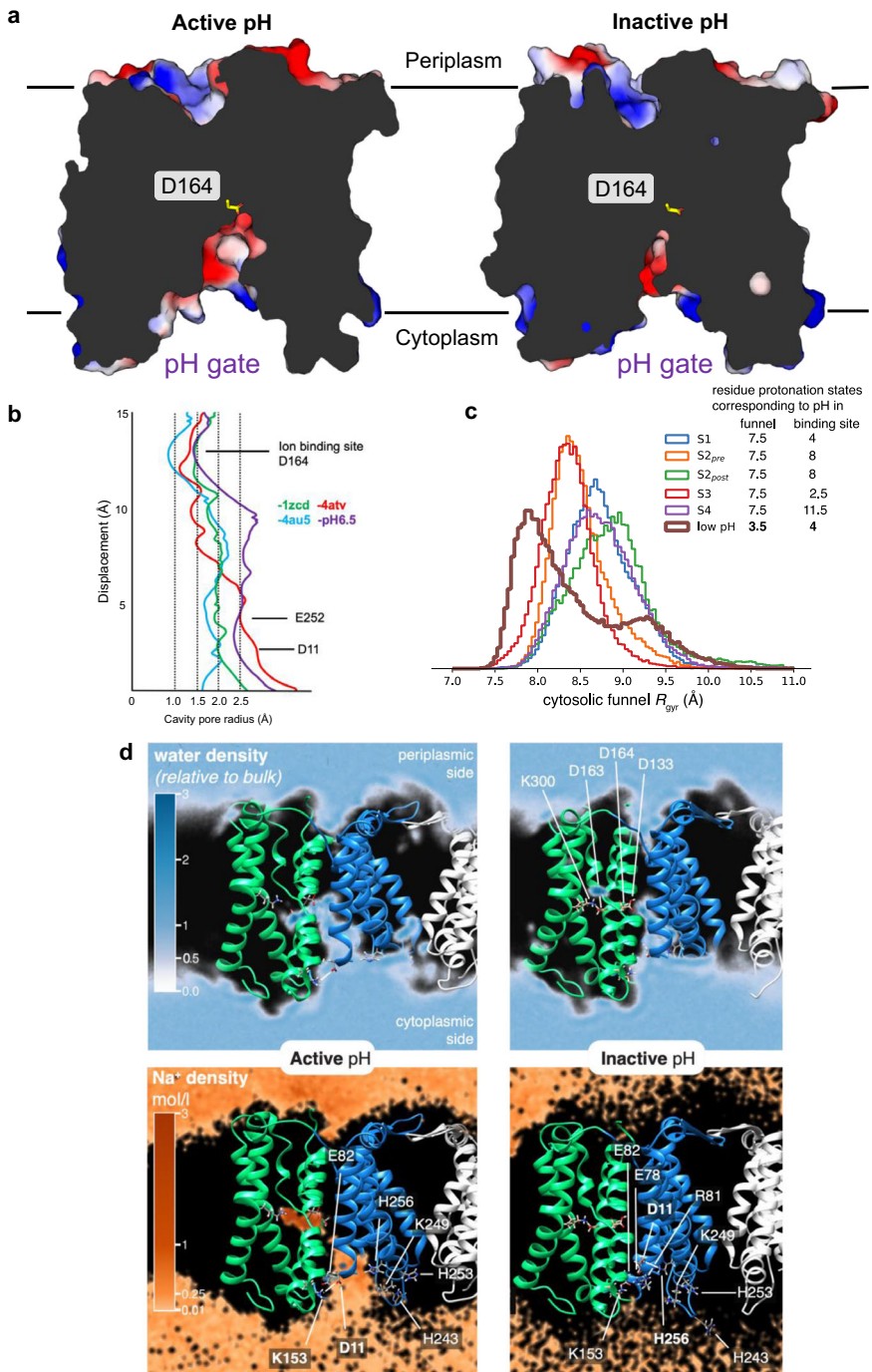

**Fig. 2 | Solvent accessibility analysis between inactive and active pH structures.** **a** Slice through an electrostatic surface representation of NhaA structures at active pH 6.5 (left, PDB ID: 7S24) and inactive pH 4.0 (right, PDB ID: 4AU5). The ion-binding aspartate is indicated and shown as yellow sticks. Notably the ion-binding funnel open to the cytoplasmic side is much more open at active pH. **b** HOLE analysis of the cytoplasmic funnel of NhaA crystal structures. **c** Opening of the cytosolic funnel in MD simulations as estimated by the distribution of the radius of gyration of the funnel-lining pH sensor residues. In the low pH simulation (brown) the funnel radius is decreased even though the MD simulation started from a high

pH structure with a more open funnel. All simulations with funnel protonation states corresponding to high (active) pH have a larger funnel radius, regardless of the protonation states of the binding site residues. **d** Density of water (blue) and Na⁺ (orange) from MD simulations under active pH (left) and inactive pH (right) conditions. Key residues in the binding site and the pH sensor are shown as sticks and labeled. The water density is measured relative to the bulk density (with value 1) while the Na⁺ density is shown in mol/l (with bulk at ~150 mM). The interaction between Lys153 and Asp11 at active pH and between His256 and Asp11 at inactive pH is indicated by a dashed line. Source data are provided as a Source Data file.

(Fig. 4b, e, f and Supplementary Fig. 8). Furthermore, Na⁺ binding in NhaA repeatedly catalyses Asp163-Lys300 salt-bridge breakage (see, e.g., Supplementary Fig. 8b, c). The simulations consistently show that when Na⁺ is bound, the salt bridge is disrupted (Fig. 4c, e, f). In particular, the salt bridge becomes weaker with decreasing number of bound protons, as indicated by a shift in the distribution of the Asp163-

Lys300 carboxylate oxygen/nitrogen amide distance from ~2.6 Å to >4 Å from the likely two-proton bound state (D164 and K300 protonated) to the zero-proton state (Asp164 and Lys300 deprotonated) (Fig. 4c). The MD simulations indicate that when Lys300 is deprotonated, the Na⁺ ion may even be directly coordinated by the lone pair electrons of the lysine ε-amino N_ζ (Fig. 4f); such a direct interaction

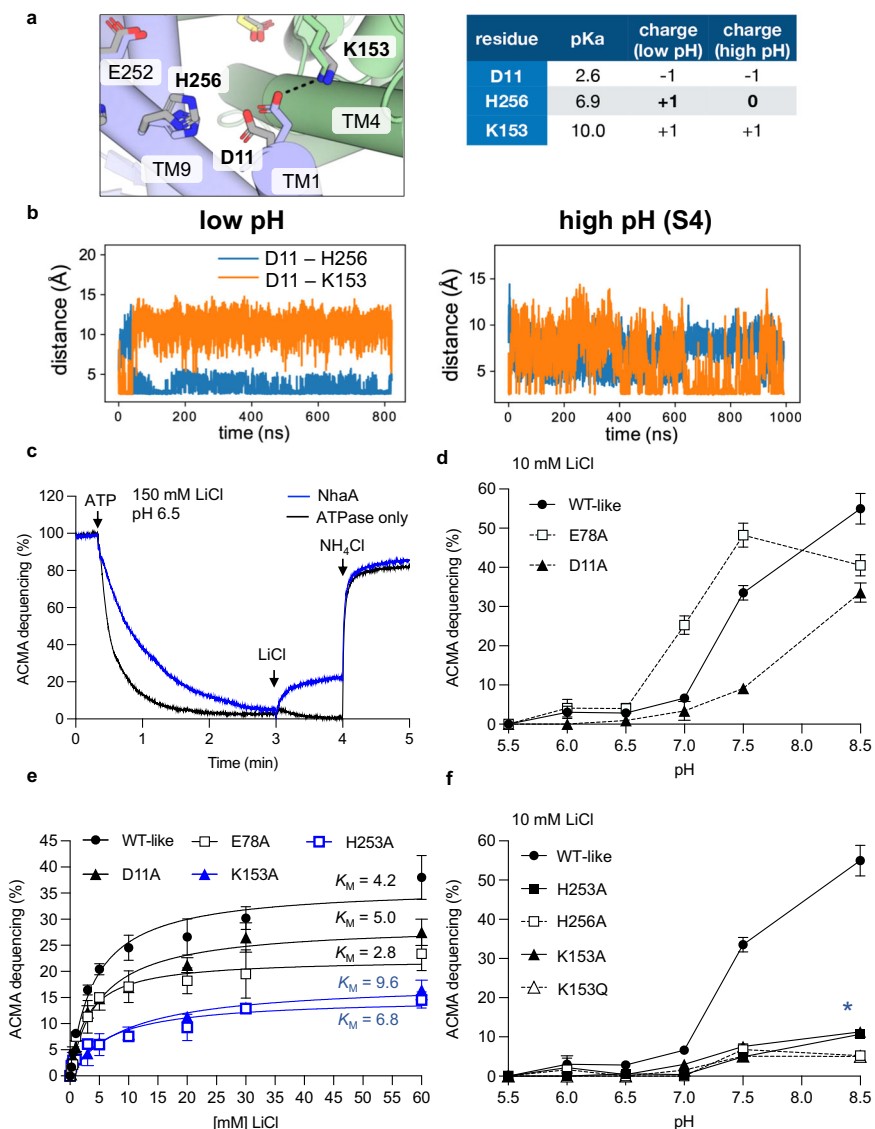

**Fig. 3 | Salt-bridge interactions in pH gate critical for pH activation. a** *left:* Salt bridge switch between D11 and H256 and between D11 and K153. *right:* Predicted p$K_a$ and charge state at low pH (3.5) and high pH (7.5). **b.** Representative MD times series of salt bridge distances for low pH (left) and high pH (right, "S4" protonation state with no protons bound in the binding site) showing salt bridge switching. The protonation state of H256 differs between low and high pH as indicated in (**a**). **c** The ATP (adenosine triphosphate) synthase and the *Ec*NhaA WT-like triple mutant were co-reconstituted in liposomes. ATP-driven proton pumping establishes a pH (acidic inside) as monitored by the quenching of 9-amino-6-chloro−2-fluorescence (ACMA). Proton efflux is initiated by the addition of 150 mM LiCl at pH 6.5 for NhaA triple mutant (blue trace) and empty liposomes (black trace). **d** pH dependence of NhaA Li$^+$-H$^+$ antiporter activity for WT-like triple mutant (closed circle), Glu78Ala (open squares) and Asp11Ala (closed triangles) were measured in proteoliposomes

by the level of ACMA dequenching as in b at the indicated pH values after the addition of saturating LiCl. The data represent means ± standard error of the mean (SEM) of *n*=3 technical repeats. **e** The NhaA WT-like triple mutant and pH-gating mutants were co-reconstituted with the ATPase in liposomes, and after the addition of increasing concentration of LiCl at pH 7.5, a kinetic curve was generated. Error bars represent SEM of *n*=3 technical repeats. The calculated $K_M$ Li$^+$ is shown next to each titration. **f** pH dependence Li$^+$-H$^+$ antiporter activity as in e. for WT-like triple mutant (filled circle), His253Ala (filled square), His256Ala (open square), Lys153Ala (filled triangle) and Lys153Ala (open triangle). Error bars represent SEM of *n*=3 technical repeats, except for His253Ala at 3 mM LiCl and Glu78Ala at 5 mM LiCl, represent an average of *n*=2 technical repeats. Asterisk (*) denotes that a zoomed in view of this figure for ACMA dequenching ranges (0 to 15%) is shown in Supplementary Fig. 7c. Source data are provided as a Source Data file.

between ion and neutral K300 completely disrupts the salt bridge (even though the Lys300 – Asp163 distance may drop below 4 Å as seen in Fig. 4c) and further stabilize the deprotonated state of the transporter. Although only a single Na$^+$ is transported at a time, in the MD simulations an additional Na$^+$ ion can occasionally associate with Asp164 as seen in Fig. 4f. Although such a spurious event indicates that the intracellular funnel and binding site at active pH constitute a favourable environment for Na$^+$ it is unclear if this observation has any physiological relevance.

Central to the transport model wherein Lys300 is the proton carrier is the breakage of the Asp163-Lys300 salt-bridge upon the

binding of Na$^+$ to enable the release of the proton from Lys300[16,40]. In previous MD simulations of *Tt*NapA, however, Na$^+$ binding did not clearly trigger breakage of the equivalent salt-bridge[18]. Furthermore, more sensitive electrophysiological experiments of an *Ec*NhaA Lys300Gln mutant could still detect electrogenic transport, albeit with 40-fold lower currents than wildtype, respectively[46]. The ion-binding sites of *Tt*NapA and *Ec*NhaA appear very similar (Supplementary Fig. 9a), but they do respond differently to mutagenesis. In particular, the Lys305Arg variant in *Tt*NapA was barely active for Li$^+$ at pH 8 ($K_M$ > 70 mM)[43], whereas an equivalent Lys300Arg variant in *Ec*NhaA retains robust activity at pH 8.5 for Li$^+$ ($K_M$ = 0.8 mM)[47]. It is plausible

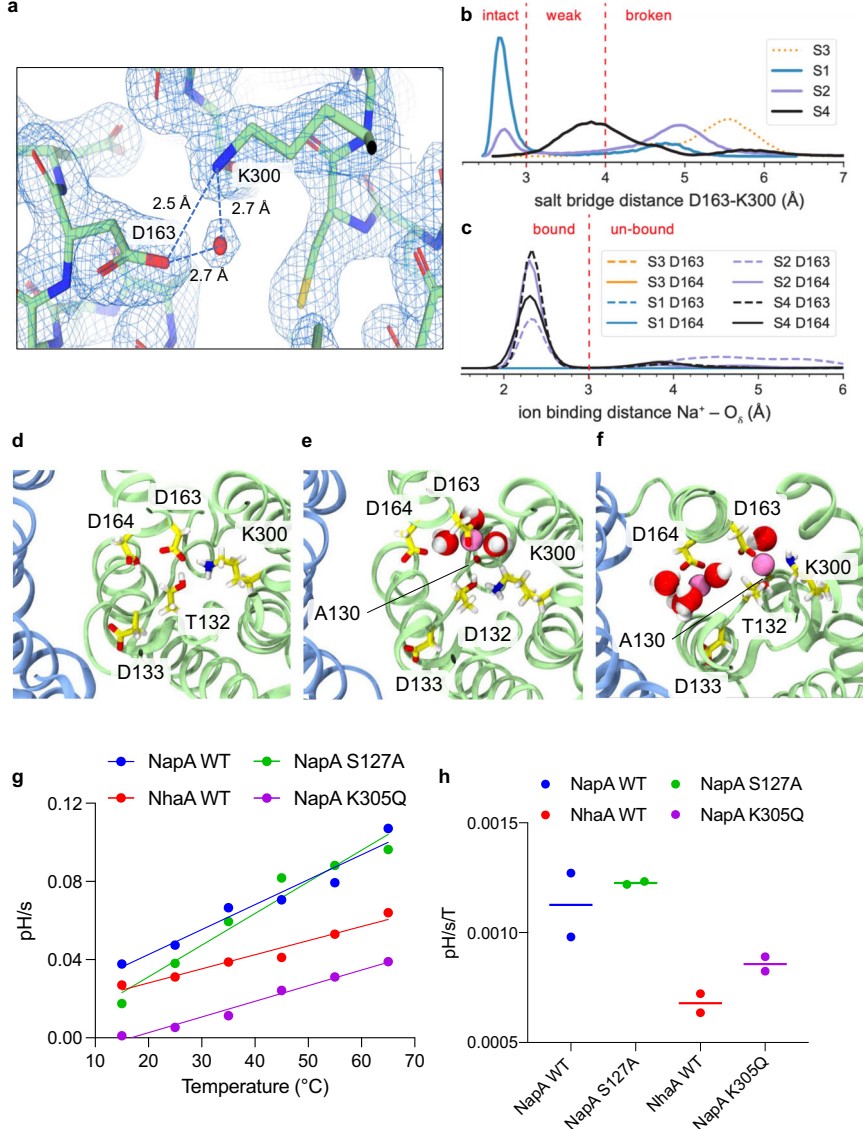

**Fig. 4 | MD simulation of ion binding and Influence of salt-bridge on temperature-dependent rates. a** The monoLCP-NhaA structure with its electron density 2Fo-Fc map at 2.5σ, showing the well resolved salt-bridge between Asp163-Lys300 and a water molecule close-by. **b** Distribution of the salt bridge distance between D163 and K300 in MD simulations at high pH, measured as the minimal distance between a carboxylate oxygen of D163 and the amide nitrogen of K300. The protonation states in the binding site were varied (S3: 3H$^+$ bound, S1: 2H$^+$ bound, S2: 1H$^+$ bound, S4: no H$^+$ bound, see Methods for details). The D163-K300 salt bridge is considered intact for distances <3 Å, weak for distances between 3 and 4 Å, and broken for >4 Å. **c** Distribution of the minimal Na$^+$ ion-carboxylate oxygen distance for the residues D163 (dashed) and D164 (solid). A Na$^+$ ion is considered bound for distances <3.5 Å. MD data for (**a**) and (**b**) were collected for each protonation state (S1–S4) and aggregated over three repeats and protomers A and B and plotted as a Gaussian kernel density estimates (KDE) with bandwidth 0.05 Å. **d** Intact salt bridge without any ion binding in state S1, the doubly-proton bound

state with both D164 and K300 protonated. (simulation S1_0 protomer B at time 350 ns). **e** Broken salt bridge with bound ion (magenta sphere) at D163 for state S2, the dominant singly protonated state where D164 has released its proton. (simulation S2_0_post protomer A at 272 ns). Water molecules in the first hydration shell (oxygen-Na$^+$ distance <3.5 Å) are shown. **f** Na$^+$-bound state S4, the doubly deprotonated state where also K300 released its proton. (simulation S4_2 protomer A at 813 ns). [S3 is a state with three bound protons (D164, D163, and K300 are protonated) that likely only occurs at pH <2.5 and is not shown.] **g** Determined initial transport rates of Napa WT and NhaA WT plotted against temperature for rates obtained with LiCl. A linear regression through these points is drawn to determine the temperature dependence of transport rates. Data presented are fluorescence ratio/min as an average of two technical repeats. **h** Slope of temperature dependence determined in (**d**) of $n = 2$ independent experiments. Source data are provided as a Source Data file.

that the strength of the equivalent Lys300-Asp163 salt-bridge is stronger in *Tt*NapA.

To test this assumption, we compared transport rates of *Tt*NapA and *Ec*NhaA in proteoliposomes entrapped with the pH sensitive dye pyranine with Li$^+$ as the substrate upon increasing temperatures 25 °C to 45 °C (Supplementary Fig. 9b–e). Over this temperature range we first confirmed that proteoliposomes containing an inactive *Tt*NapA mutant had no measurable proton leakage (Supplementary Fig. 9c).

When transport was driven by either a membrane potential (ΔΨ) of −116 mV or by an ΔpLi$^+$ gradient, a greater temperature dependence was apparent for *Tt*NapA over *Ec*NhaA (Supplementary Fig. 9d, e, Fig. 4g, h). We next disrupted the salt-bridge in *Tt*NapA with an Ly305Gln variant, which has a similar $K_M$ as wildtype for Li$^{+43}$. The underlying assumption is that if breakage of the salt-bridge would be significant barrier to *Tt*NapA activity, then the Lys305Gln mutant would show less temperature dependence rates than wildtype. Indeed,

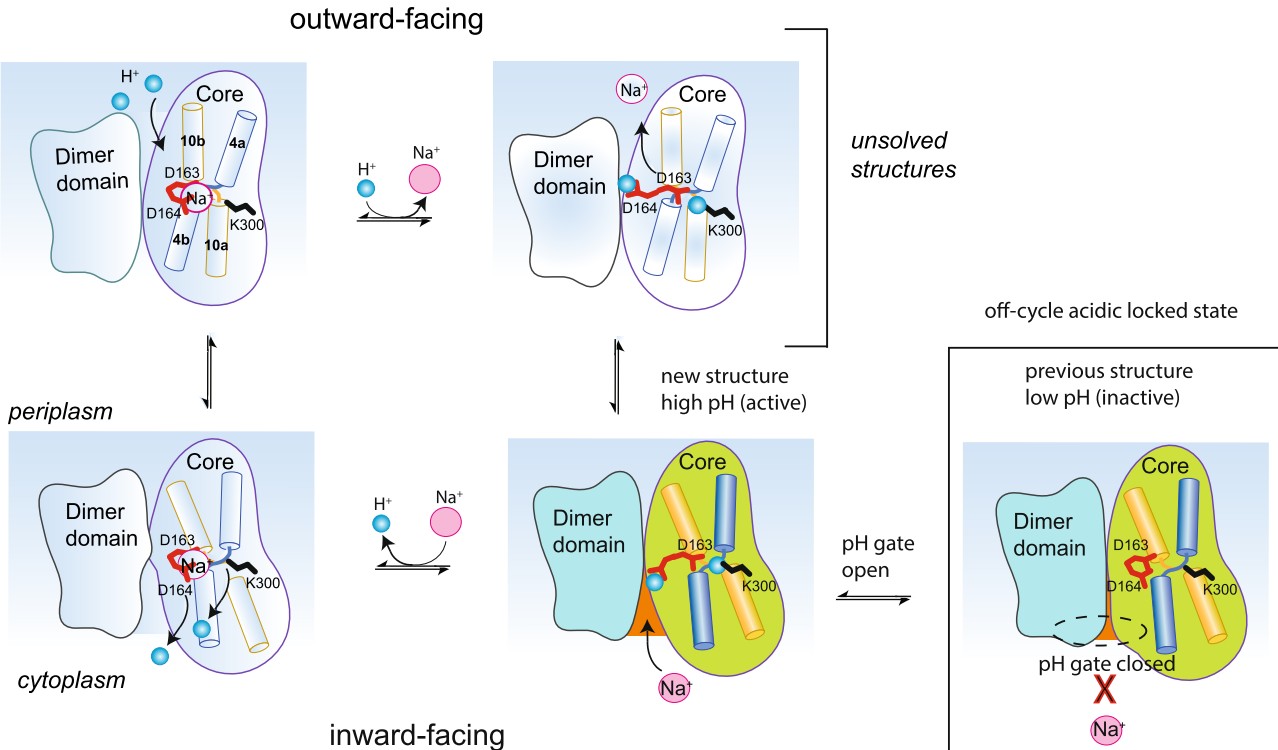

**Fig. 5 | Schematic of the incorporation of the pH gating model with the alternating-access mechanism.** The previous NhaA structures determined at low pH (PDB ID: 4AU5, 1ZCD) have been in acidic-locked conformation (<pH 4), which we interpret as an off-cycle intermediate unable to bind substrate Na[+] as the pH gate is closed (NhaA coloured in blue/green and outlined with box). The NhaA structure at active pH presented here (NhaA coloured in blue/green) has an open cavity caused by a rearrangement of charged residues involving histidine residues located in TM9. The active pH structure is part of the transport cycle and at active pH the binding of Na[+] from the cytoplasmic side to D163 and D163 triggers breakage of the D163-K300 salt-bridge and release of protons and catalyzes rearrangement to the outward-facing state. In the outward-facing state, Na[+] is released and D164 and D163 residues are immediately re-protonated in the periplasm due to the inwardly-directed proton gradient. Notably, although the transport cycle of NhaA at active pH can be described as a simple competition for Na[+] vs H[+], the pH gating residues controls the pH for which the ion-binding site becomes accessible to its substrate. In this way, modifying NhaA residues in the pH gate, some - 9 Å from the ion-binding site can control the pH at which NhaA becomes active. This channel-like activation mechanism could be present in other ion-transporters.

the Lys305Gln variant shows less temperature dependence, with temperature dependence rates similar to *Ec*NhaA (Fig. 4g, h and Supplementary Fig. 10). Lastly, to confirm this observation was not an indirect effect, due to disrupting an interaction close to the ion-binding site, we also re-measured the rates of Ser127Ala, which is in the vicinity of the ion-binding site, but not thought to coordinate Na[+] and its mutation to alanine has a similar $K_M$ for Li[+] as the Lys305Gln mutant[43]. The Ser127Ala mutant displayed the same temperature dependence rates as *Tt*NapA wildtype (Fig. 4g, h and Supplementary Fig. 10).

## Discussion

Intracellular pH must be exquisitely regulated to maintain cell homeostasis and Na[+]/H[+] exchangers are key proteins required to fine-tune intracellular and organellar pH[2,4]. The pH dependence activity of NhaA is likely connected with its role in pH homeostasis under different environmental stresses, and the bacterium's adaptation to high salinity[39]. The mechanism of pH activation of NhaA has spanned decades of research, leading to the identification a charged network on the cytoplasmic side that controls the pH for activation[3,48]. Nevertheless, it has been unclear how to interpret the pH sensor model when the pH dependency of NhaA fits a simple H[+] *vs* Na[+] (Li[+]) competition model at active pH[34].

In this study, we can demonstrate for the first time, that between inactive pH 4 and active pH 6.5, the charged network on the cytoplasmic side of NhaA rearranges to open up the ion-binding site to the cytoplasm. MD simulations and HOLE analysis support the interpretation that the rearrangement of the charged network is required to enable the passage of a hydrated Na[+] ion; at pH 4.0 the pathway is too narrow and electrostatically unfavourable. MD simulations further highlight how the protonation state of the ion-binding site itself does not influence the size of the cytoplasmic funnel, only the external pH. Indeed, starting with the high-resolution NhaA structure at active pH, the cytoplasmic funnel closes over the course of the MD simulations when the external pH is lowered, due to rearrangements of the charged network. Structural analysis, MD simulations and functional analysis of pH gate mutants, highlight the critical role of His253 and His256 residues in the dimer domain and Lys153 in the core domain in pH activation. We propose that rather than the $pK_a$ of the aspartate residues in the ion-binding site, the $pK_a$ of the histidine residues in the charged network is dictating the activation pH under normal conditions. Thus, although the charged network is acting to sense cytoplasmic pH, its functional role is more akin to a pH gate, controlling accessibility to the ion-binding site, more consistent with earlier MD simulations[40].

We propose the pH gating mechanism of NhaA is part of a separate, off-cycle intermediate in the transport cycle, and is somewhat analogous to the opening of a pH-gated ion-channel (Fig. 5). However, instead of the ions directly moving down their electrochemical gradient and across the membrane, they bind to the central substrate-binding site and are transported by an elevator alternating-access mechanism, driven by H[+] *vs* Na[+] binding[34]. Elevator proteins are typically oligomers[49] and the dimeric form of NhaA is stabilised by cardiolipin[35,37,38]. The arginine residues Arg203 and Arg204 that are

thought to coordinate cardiolipin in the dimerization interface[35,38], as supported here by thermal-shift assays (Supplementary Fig. 2b), are located proximal to the pH gating region (Supplementary Fig. 2c). It is thus possible that cardiolipin binding stabilises the dimer to further facilitate opening of the pH gate, as we previously postulated[38]. Consistently, Arg203Ala and Arg204Ala variants shift the pH for activation from pH 6.5 to pH 7 to 7.5[38], and the addition of cardiolipin to detergent purified NhaA increases the apparent binding affinity of $^{22}$Na[42].

It seems likely that the herein described pH gating mechanism would also be utilised by other pH-dependent transporter systems. Indeed, the mammalian Na$^+$/H$^+$ exchanger structure NHE9[50] has revealed clusters of well-conserved histidine residues at the cytoplasmic cavity entrance (Supplementary Fig. 6g), opening up a new avenue to explore in pH-sensing in light of our analysis. Lastly, MD simulations of the high-resolution NhaA active state structure show that once Na$^+$ is able to access the ion-binding site, the salt-bridge between one of the ion-binding site residues and a lysine residue breaks to bind Na$^+$. Consistent with this model, upon the addition of increasing concentrations of Na$^+$ in 2D crystals of MjNhaP, local rearrangements were only observed in the core domain[51]. Moreover, crystallographic water molecules are located next to Lys305/Lys300 in both high-resolution structures of TtNapA[18] and EcNhaA, which further implies that the lysine residue would be hydrated sufficiently to break a favourable interaction to the aspartate upon Na$^+$ binding. Interestingly, salt-bridge breakage can be a significant activation barrier at room temperature, as seen in thermophilic TtNapA, where removal of the salt-bridge gave rise to a protein showing similar temperature dependent rates as EcNhaA. It is possible that these differences can be further attributed to adaptation to different requirements of the host organism. E. coli is a mesophile with optimum growth at 37 °C present in the human gut, whereas T. thermophilus is a thermophile, preferentially growing around 65 °C in the thermal vent environments of hot springs.

In summary, our data reveals the structural basis for pH activation in EcNhaA and shows that an Na$^+$ salt-bridge breakage is important next step in the transport cycle. As far as we are aware, this is the first example seen in a secondary-active transporter where activity is controlled by intrinsically modulating accessibility to the substrate-binding site, and might represent a form of allosteric regulation used in other type of ion-transporters in particular.

## Methods

### Expression and purification of recombinant TtNapA and EcNhaA
EcNhaA WT-like triple mutant (A109T, Q277G, L296M), TtNapA wild type and mutants were previously cloned into the IPTG inducible expression vector pWaldo-GFPe with a TEV-cleavable C-terminal GFP-His$_8$ tag were overexpressed in the E. coli strain Lemo21 (DE3) as previously described[16]. The TtNapA triple mutant, TtNapA wildtype and mutants thereof were extracted from membranes with n-Dodecyl β-D-maltoside (DDM; Glycon) and purified by Ni-nitrilotriacetic acid (Ni-NTA; Qiagen) affinity chromatography, GFP-His$_8$ TEV cleavage, and size-exclusion chromatography as previously described[16].

### Crystalization of EcNhaA WT-like triple mutant by LCP
The EcNhaA WT-like triple mutant was purified with buffers containing of 2% (w/v) cardiolipin (18:1) (Avanti, cat. no. 710335 P), concentrated to 36 mg.ml$^{-1}$ and used for in meso or lipidic cubic phase (LCP) crystallization. The purified GFP free EcNhaA (36 mg.ml$^{-1}$) was mixed with molten monoolein (Sigma, CAS Number 111-03-5) in a weight ratio of 2:3 respectively, using coupled syringe-mixing device (Hamilton). A transparent cubic phase was formed and crystallization trials were set up by dispensing 50 nL cubic phase onto a 96-well Laminex glass plate (MD11-50, Molecular Dimensions), which was then covered with 800 nL of crystallization solution (0.1 M MES pH 6.5, 0.1 M NaCl, 0.1 M CaCl2, 24–45% (v/v) PEG 400), with an LCP Mosquito robot (TTP

Labtech). Plates were sealed with a Laminex glass cover (MD11-52, Molecular Dimensions) and were stored at 20 °C. The LCP crystals obtained were harvested without any cryo-protectant and cryocooled in liquid nitrogen.

### X-ray data collection and refinement of EcNhaA
The diffraction data was collected at ESRF ID30A-3 on an Eiger X 4 M detector with 0.968 Å wavelength X-rays with flux 8.65 x 10$^{11}$ph s$^{-1}$, with 1° oscillation and 0.05 s exposure per frame. Data was processed using the XDS package[52] and MOLREP[53] was used to obtain phase estimates in the CCP4 suite using the coordinates from our previously published dimer-EcNhaA (PDB ID: 4AU5) as a search model. The model was iteratively rebuilt using Coot[54], interspaced with refinement using Refmac[55] and Phenix[56].

### Functional characterization of pH gating residues
The EcNhaA WT-like triple protein and variants of this construct containing individual pH gating mutants, were purified as GFP-fusions (no TEV cleavage) and co-reconstituted with purified F-type ATP synthase from E. coli with a 2:1 molar ratio (NhaA/ATP synthase) in MME buffer (10 mM MOPS-NaOH, pH 8.5, 2.5 mM MgCl2, and 100 mM KCl) as described previously[17,43]. Typically, 50 μl proteoliposomes were added into 1.5 ml of MME buffer containing 3 nM 9-aino-6-chloro-2-methoxyacridine (ACMA) and 140 nM valinomycin. Fluorescence was monitored at 480 nm at an excitation wavelength of 410 nm using a fluorescence spectrophotometer (Cary Eclipse; Agilent Technologies). An outward-directed pH gradient (acidic inside) was established by the addition of 2 mM ATP, as followed by a change in ACMA fluorescence. After a 3-min equilibration, the activity of the EcNhaA triple mutant and variants was assessed by the dequenching of ACMA fluorescence, resulting from the addition of LiCl. A further addition of 20 mM NH$_4$Cl was further added after 1 min to obtain nearly complete ACMA dequenching. By measuring ACMA dequenching by the addition of 10 mM LiCl at pH 5.5, 6.0, 6.5, 7.0, 7.5, 8.5, respectively, the effect of pH on EcNhaA triple mutant activity was assessed. For calculation of an apparent $K_M$, the above experiment was repeated at pH 7.5 and with the addition of increasing concentrations of LiCl (0 to 60 mM). The fraction of ACMA dequenching as a function of LiCl concentration were fitted to Michaelis–Menten kinetics using nonlinear regression by GraphPad Prism 7.0. Each experiment was performed in triplicate.

### Transport rate measurement of TtNapA and EcNhaA with liposome entrapped pyranine
Proteoliposomes of TtNapA variants or EcNhaA were prepared as described previously[17]. In brief, for ΔΨ-driven transport, liposomes prepared in 2 mM MOPS-BisTris, pH 7.8, 50 mM K$_2$SO$_4$, 50 mM Li$_2$SO$_4$, 1 mM pyranine were diluted into 2 mM MOPS-BisTris, pH 7.8, 50 mM Li$_2$SO$_4$, 0.5 mM K$_2$SO$_4$, and a membrane potential of −116 mV was induced by addition of 200 nM valinomycin. For substrate driven transport, liposomes prepared in 2 mM MOPS-Tris, pH 7.8, 50 mM K$_2$SO$_4$, 0.5 mM Li$_2$SO$_4$, 0.5 mM Na$_2$SO$_4$, 1 mM pyranine were diluted in the same buffer containing 200 nM valinomycin, but lacking pyranine. Proton efflux was induced by addition of saturation concentration of 40 mM substrate (NaCl/LiCl). In both experiments, the change in pH was monitored by measuring the ratio of pyranine fluorescence at 406 nm$_{ex}$/510 nm$_{em}$ against 460 nm$_{ex}$/510 nm$_{ex}$. The pH at every time point was calculated with a pH calibration curve of pyranine as described previously[43]. To determine the transport rate, a linear regression was performed on data collected during the first 4 s after the substrate induced response and the slope calculated as ΔpH/s.

### Thermostabilization by cardiolipin measurements of EcNhaA GFP-fusion mutants
To characterize the thermostability by cardiolipin of an EcNhaA - Arg203Ala Arg204Ala double-mutant we utilized the previously

described GFP-TS assays[35]. In brief, purified *Ec*NhaA Arg$_2$O$_3$Ala-Arg$_2$O$_4$Ala GFP-fusion was diluted to a final concentration of 0.05–0.075 mg/ml and incubated in the presence of 1% (w/v) DDM for 30 min at 4 °C in conditions (i) no lipid added, and (ii) 20 mg/ml cardiolipin added. Subsequently β-D-Octyl glucoside (Anatrace) was added to a final concentration of 1% (w/v) and the sample aliquots of 100 μl were heated for 10 min over a temperature range of 20–70 °C in a PCR thermocycler (Veriti, Applied Biosystems) and heat denatured material pelleted at 18,000 × *g* during 30 min at 4 °C. The resulting supernatants were transferred to a 96-well plate and the fluorescence determined with a plate reader (Thermo-Scientific). The apparent $T_m$ was calculated by plotting the average GFP fluorescence intensity from three technical repeats per temperature and fitting the curves to a sigmoidal 4-parameter logistic regression in GraphPad Prism software.

### Molecular dynamics simulations

The initial system for the simulations of the NhaA dimer was constructed by superimposing the new monomeric pH 6.5 structure on the crystallographic dimer of the 2014 structure (4ATV[16]). The dimer was embedded in a 4:1 POPE:POPG (1-palmitoyl-2-oleoyl-sn-glycero-3-phosphoethanolamine, 1-palmitoyl-2-oleoyl-sn-glycero-3-phosphoglycerol) membrane with a free NaCl concentration of approximately 150 mM using CHARMM-GUI[57–59] and simulated with the CHARMM36 force field for proteins and lipids[60,61] and the CHARMM TIP3P water model. Protonation states of ionizable residues were selected based on the p$K_a$ values determined by constant pH MD simulations[40] for a nominal pH of 7–8 in the following manner unless mentioned otherwise. Protonation states of key residues Asp163, Asp164, and Lys300 were fixed (see[16] for details). The "salt bridge mechanism" hypothesis progresses through states S1 (two protons bound to D164 and K300), S2 (one proton bound to K300), S4 (no protons bound), with Na$^+$ binding in state S2 and S4 where the Asp163-Lys300 salt bridge is destabilized by the bound ion and Asp164 and Lys300 function as the proton carriers[16,40]. The S3 state has all three residues protonated, as would be expected for a "two Aspartate" mechanism in which Asp163 and Asp164 carry the protons. In simulations S1–S4 we kept Asp133 deprotonated and the cytosolic funnel is simulated with protonation states appropriate for -pH 7.5 while the protonation states of the binding site residues effectively correspond to pH 2.5 (S3), pH 4 (S1), pH 8 (S2), and pH 11.5 (S4)[40]. Additionally, a simulation at nominal pH 3.5 (named "low pH") was performed during which all residues were set according to the pKa values in[40], which means in particular that the binding site residues were kept in the S1 state and that Asp133 was protonated[45].

Each system was energy minimized and equilibrated under restraints following the CHARMM-GUI protocol[57]. Three MD simulations in the *NPT* ensemble at standard conditions were performed for each set of protonation states, ranging from 0.58 μs to 1 μs in duration (Supplementary Table 2). All simulations started with no Na$^+$ ions near the binding site but in both S2 and S4 simulations, ions spontaneously entered the cytosolic funnel and bound to Asp164/Asp163 during the initial equilibration phase. In order to assess spontaneous binding in the S2 state independent of the initial equilibration phase, these simulations were stopped after ~0.6 μs (simulations *S2_0-pre, S2_1-pre, S2_2-pre* in Supplementary Table 2), the bound ion exchanged with a bulk water molecule, and then continued (simulations *S2_0-post, S2_1-post, S2_2-post* in Supplementary Table 2). In all three "post" simulations, another Na$^+$ ion bound again to Asp164 and Asp163, showing that binding is spontaneous and rapid.

Simulations were performed with GROMACS 2018[62] with other details as for our previous simulations of NapA[18]. Because versions of GROMACS < 2018.6 contained a bug (https://gitlab.com/gromacs/gromacs/-/issues/2845) that could lead to unstable CHARMM36 membrane protein simulations we checked that in our simulations the bug was not triggered, as indicated by stable box dimensions with only small fluctuations and the protein not diffusing across the periodic boundaries of the simulation cell. Simulations were analyzed with Python code based on MDAnalysis[63] (distances, radius of gyration, density) and GROMACS tools[62] (DSSP secondary structure analysis). The opening of the funnel was assessed by calculating the radius of gyration $R_g = \sqrt{\dfrac{\sum_{i=1}^{N} m_i \Delta r_i^2}{\sum_{i=1}^{N} m_i}}$ of the inner funnel lining pH sensor residues Asp11, Glu78, Arg81, Glu82, Lys153, Glu252, His253, His256 and the distribution of $R_g$ was calculated by histogramming the time series over all repeats and protomers of the simulations with the same protonation states. The time series of the distance between the amide N$_\zeta$ of Lys300 and the closest carboxylate O$_\delta$ of Asp163 was extracted from the frames of the simulation. The salt bridge was considered intact if the Asp163-Lys300 distance was ≤3 Å (where a typical tight salt bridge -2.5 Å), weak for >3 Å and ≤4 Å and broken if > 4 Å. To assess Na$^+$ binding, the distance of any Na$^+$ ion to the closest carboxylate oxygen of either Asp163 or Asp164 was calculated. An Na$^+$ ion was considered bound for any distances <3 Å. Na$^+$ and water densities were histogrammed on a 1 Å and 0.5 Å grid, respectively, using trajectories with all frames superimposed while minimizing the C$_\alpha$ RMSD of protomer A with those of the reference structure, protomer A of PDB 4ATV. The low pH ("inactive pH") densities for both water and sodium were calculated over three independent *low pH* simulations (lowpH_1, lowpH_2, lowpH_3 in Supplementary Table 2) containing a total 24,500 frames over 2.45 μs of simulation time. The high pH ("active pH") densities for both water and sodium were calculated over three independent *S4* simulations (S4_1, S4_2, S4_3 in Supplementary Table 2) containing a total of 30,000 frames over 3 μs of simulation time.

### Molecular visualization

Figures were generated in PyMol[64] and VMD[65]. Densities from MD simulations were visualized with the density analysis tools[66] in Chimera[67].

### Reporting summary

Further information on research design is available in the Nature Research Reporting Summary linked to this article.

## Data availability

The coordinates and the structure factors for *Ec*NhaA LCP have been deposited in the Protein Data Bank under accession ID 7S24. Source data are provided with this paper.

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

## Acknowledgements

LCP crystals were harvested and screened at ESRF by Emmanuel Nji with excellent assistance from beamline scientists. We thank Leticia Herrán Villalaín and Philipp Müller (University of Bern) for providing purified ATP synthase from *E. coli*. Vetenskapsrådet initially funded this work to (DD), with continued support from the European Research Council (ERC) Consolidator Grant EXCHANGE (Grant no. ERC-CoG-820187) to D.D. Research reported in this publication was supported by the National Institute of General Medical Sciences of the National Institutes of Health under Award Number R01GM118772 to O.B. Computing time on the Agave cluster of Research Computing at Arizona State University is gratefully acknowledged.

## Author contributions

D.D. designed and supervised the project. Structural experiments were performed by P.U., J.B., R.M. and D.D. Biochemical experiments were performed and analysed by I.W., P.U., L.W., F.G., P.M., S.J., C.vB., and D.D. Molecular dynamics simulations were performed by I.M.K. and analyzed by I.M.K and O.B. The manuscript was prepared by D.D., O.B., and I.W. with contributions from all authors.

## Funding

## Competing interests

The authors declare no competing interests.
