## [Peer Review File · Nature Communications]

Crystal structure of the Na⁺/H⁺ antiporter NhaA at active pH reveals the mechanistic basis for pH sensing

Editorial Note: Parts of this Peer Review File have been redacted as indicated to remove third party material where no permission to publish were obtained.REVIEWER COMMENTS

Reviewer #1 (Remarks to the Author):

Na⁺/H⁺ antiporters transport sodium in exchange for H⁺. They play a role in the regulation of pH and sodium levels. The most extensively studied Na⁺/H⁺ antiporter is Ec-NhaA, the main Na⁺/H⁺ antiporter of *Escherichia coli*. Ec-NhaA activity is highly pH-dependent, inactive at pH 4, and rates increase 2000-fold from pH 6.5 to pH 8. The first EcNhaA crystal structure was determined at 3.45 Å at pH 4. The new structure presented by Winkelmann and collaborators was determined at pH 6.5 at 2.2 Å. The authors show that the pH 6.5 EcNhaA undergoes a channel-like opening of the intracellular funnel to enable Na⁺ accessibility to the ion-binding aspartate Asp164.

It is very disconcerting to read throughout the paper that this conformational change, albeit interesting, is the structural basis for pH regulation. All the available experimental evidence cited by the authors demonstrates that the antiporter is inactive at pH 6.5 yet, they coin the pH 6.5 structure the active one (not once but 25 times throughout the paper), and 11 times they state that 6.5 is the activating pH. In the discussion, this reviewer was further confused when they wrote that NhaA is 10% active at pH 6.5 while they previously stated that activity increases 2000 fold between 6.5 and 8.0. To add to the confusion, they cite Na-binding assays that already show half-maximal binding of sodium to NhaA at pH 6.0. How does fit the contention that the channel-like opening enables Na⁺ accessibility? Actually, and even more confusing, in the cited reference, binding at pH 4 was the same as at pH 6.

The new structure provides some interesting information. However, it may or may not be relevant to the topic of pH regulation.

Minor comments:

- 1) In the model in Fig.1, the cytoplasm is at the top while, in all the others, it is at the bottom. In Fig. 2, they used "Out and In" for cytoplasm and periplasm.
- 2) From the Introduction: "... bacterial homologues are often" Can the bacterial antiporters be considered homologs of the human? They catalyze the same reaction, but they are very different functionally (electrogenicity and pH activation), and they share minimal sequence similarity, if at all.
- 3) Some editing and spell-checking are necessary.

Reviewer #2 (Remarks to the Author):

The manuscript by Winkelmann et al. reports the X-ray crystal structure of the E. coli sodium proton antiporter NhaA at pH 6.5. The structure reveals novel details concerning the mechanism for pH activation in this system through structural comparisons with the inactive state of the same protein, determined at pH 4.0 (PDB: 4AU5). Accompanying detailed MD and biochemical assays further support the main novelty in the study that the main driver for pH activation is that protonation of residues in the pH sensor results in the opening of a channel that allows hydrated sodium to access the central ion binding site. This model supports previous work on this and related systems (all referenced in the study) that identified the histidines on TM9 as likely to be the physiological proton binding sites. Additionally, the study then explores the effect of pH on the mechanism of transport as it relates to the Asp164-Lys300 salt bridge. Here the authors use their higher resolution structure at pH 6.5 to further detail the steps surrounding the breakage of this interaction following sodium binding to Asp163.

Overall, I found the present study has some interesting new information that does significantly add to the current literature on sodium proton antiporters, most notably the visualisation of the effect of pH activation. However, I also found the paper hard to read and somewhat disjointed between the two sections. At times I was also unsure what was novel and what was already known in the literature.

I very much liked the first section, which compares the structure of the activated vs inactive transporter (Fig. 1). However, the description lacked insight into the mechanism of activation. I understood that protonation results in the reorientation of the salt bridges from interhelical to intrahelical and the importance of the His256-Asp11 interaction. But I was left wondering whether the authors had pinpointed the protonated residues or worked out, via their MD analysis, the actual steps in the activation process. I assume that protonation of either His253 or His256 is the first event in the activation process, but I couldn't determine whether the MD showed this? Without this, it appeared to me that the current structure essentially supports the previous literature, which had already identified roughly where the pH sensor was located but hadn't to date recorded the steps involved.

I wondered whether the authors could use their all-atom MD simulations, detailed on page 8, to replicate the activation process, at least locally, and observe the structural rearrangements displayed in their structural comparison. Could a systematic protonation of the residues and resulting analysis identify the key sites on proton binding? These results may enable the authors to pinpoint a primary protonation site for activation or whether the proton can bind to several places and still result in activation.

The authors compare their new structure at pH 6.5 to their previous structure at pH 3.8. However, the structure at pH 3.8 was determined at 3.7 angstroms. Given the relatively minor changes in side-chain positions being discussed, how confident are the authors of the side chain locations in their previous structure?

Page 8 – the author's highlight that Glu252 changes side-chain conformations in line with pH activation. However, as drawn in Fig. 1C, the side chain does not appear to move very much at all?

Page 9 – details the second part of the study, which focuses on the Asp164-Lys300 salt bridge. The authors focused on this interaction pertaining to the mechanism of sodium and proton binding in the central ion binding site.

I was very confused by this section, and I didn't understand what the authors had discovered. How does the temperature-dependent rate analysis link with the pH activation of the proton sensor? This section appeared to be included to explain the differences observed in previous papers of the effect of mutating Lys300 to Arginine, which differs between *E. coli* Nha and *T. thermophilus* NapA (Lys305), but why? It was also unclear how this explained the observation that a membrane potential could no longer drive transport via the Lys305Gln mutant. Is this linked to pH activation or the pH? The last paragraph of this section also seemed separate from the previous paragraphs. Again, I was unsure how measuring the melting temperatures resulted in further insight into pH activation, which is the paper's title?

Finally, I was not convinced by the argument in the concluding paragraph that sodium proton antiporters have a unique mechanism for activation that involves regulating access to the central binding site. As described, the gate that allows accessibility would need to be separate from the transport domain, such that you could mechanistically separate the two functions. From reading this study, it seems to me that the pH sensor has to be protonated to release the salt bridge locks that enable the transporter to move. This seems more similar to the situation with the glutamate transporters, where the rate-determining step is the unlocking of the interactions between the bundle and scaffold domains. Here the rate-determining step is the protonation of the pH sensor, which unlocks the transport for activity.

Minor points:

Fig.1. The transporter is shown in the LacY orientation (cyto up) but in Fig. 2 (cyto down). I would correct this.

I would also check the consistency of labelling. As above, Fig. 1 (cyto/periplasm) in Fig. 2 (In/Out). Fig. 1C, TM5 is TMV. Fig 1D, helices need labelling and the sodium tunnel labelled.

Reviewer #3 (Remarks to the Author):

The manuscript by Winkelmann et al presents crystal structure of the Na/H antiporter NhaA at activating pH condition. Combining this structural model with molecular dynamics (MD) simulations, the authors address the mechanistic basis for pH sensitivity in this system.

While the topic is very interesting, I find that presentation of the manuscript is not the most optimum, especially when it comes to description of the MD methods and results. Therefore, in the current form of the manuscript, I cannot properly evaluate the manuscript. Below I list several key aspects of the presentation that I think the authors should address to bring more clarity to their work:

1. I find it remarkable that only in the Methods section the authors describe so called “salt bridge mechanism” and “two Aspartate mechanism” and based on these mechanisms they present classification of states they simulate (all this in Methods only). This information is so buried in the manuscript that it is pretty much impossible to follow which states and mechanisms are probed in the simulations as they are presented in the main text. These mechanisms and the associated states (S1-S4) must be presented either in the Intro or in the Results, and then when the MD data is described, each simulation condition must be associated with one of the states in the text. As it stands, the manuscript talks about high/low pH conditions in Results, S1-S4 states are defined in the Methods (and never used in the Results) and there is no connection between them (at least for a person who does not work on NhaA transporter).

2. The assignment of the protonation states is presented in an equally confusing way. I would recommend adding to Table S3 protonation states for each of the key residues. I do realize that the manuscript refers several times to the earlier publications on this topic (as well as for S1-S4 state

definitions), but still, I believe that such fundamental mechanistic concepts should be re-emphasized in this manuscript as well. This will also help to determine the novelty of this work.

There are also several statements in the paper that requires clarification or revision:

1. The manuscript states (on page 7) that MD simulations display little conformational drift. But some RMSD traces in Supplemental figures 2 and 3 look not at all converged. So, I am not sure whether those simulations can be considered as stable/converged.
2. On page 6, when discussing cardiolipin binding residues, the manuscript states that their data on R204/R205 mutants “is consistent with in vivo data that showed these residues were the most important for homodimerization and function”. To me, this is a pretty big leap. I do realize that there is an expectation that those Arg residues can bind cardiolipin and previously resolved dimer construct had a lipid-like density at the dimer interface which was attributed to cardiolipin. But is there any data (especially in vivo) which shows that cardiolipin is important for dimerization and function?
3. On page 7, sentence “The loss of the TM2-TM9 interaction has likely caused a rearrangement of Lys249, His253 and His256 in TM9” is based on comparison of two frozen structures. It is hard to determine “causality” from such comparison. It is best to limit description to observed differences between the two structures.

Crystal structure of the Na⁺/H⁺ antiporter NhaA at active pH reveals the mechanistic basis for pH sensing

Corresponding authors:
Oliver Beckstein
David Drew

We appreciate the positive response concerning our manuscript. We have carefully examined each remark and responded to all points below.

REVIEWER COMMENTS

Reviewer #1 (Remarks to the Author):

Na⁺/H⁺ antiporters transport sodium in exchange for H⁺. They play a role in the regulation of pH and sodium levels. The most extensively studied Na⁺/H⁺ antiporter is Ec-NhaA, the main Na⁺/H⁺ antiporter of Escherichia coli. Ec-NhaA activity is highly pH-dependent, inactive at pH 4, and rates increase 2000-fold from pH 6.5 to pH 8. The first EcNhaA crystal structure was determined at 3.45 Å at pH 4. The new structure presented by Winkelmann and collaborators was determined at pH 6.5 at 2.2 Å. The authors show that the pH 6.5 EcNhaA undergoes a channel-like opening of the intracellular funnel to enable Na⁺ accessibility to the ion-binding aspartate Asp164.

It is very disconcerting to read throughout the paper that this conformational change, albeit interesting, is the structural basis for pH regulation. All the available experimental evidence cited by the authors demonstrates that the antiporter is inactive at pH 6.5 yet, they coin the pH 6.5 structure the active one (not once but 25 times throughout the paper), and 11 times they state that 6.5 is the activating pH. In the discussion, this reviewer was further confused when they wrote that NhaA is 10% active at pH 6.5 while they previously stated that activity increases 2000 fold between 6.5 and 8.0. To add to the confusion, they cite Na-binding assays that already show half-maximal binding of sodium to NhaA at pH 6.0. How does fit the contention that the channel-like opening enables Na⁺ accessibility? Actually, and even more confusing, in the cited reference, binding at pH 4 was the same as at pH 6. The new structure provides some interesting information. However, it may or may not be relevant to the topic of pH regulation.

Thank you for comments. In hindsight, we should have just used the term “active pH” rather than “activating” pH. Based on steady-state transport measurements, activity is measurable for WT above pH 6.5. Above this pH, the transport model is described as simple competition between Na⁺ vs H⁺, which is thought to be solely described by the pKa of the ion-binding site. However, there is a large body of data spanning 10 years

on the pH sensor of NhaA, which is a charged network located on the surface that is thought to decide “when” NhaA becomes active. How does this charged network connect to ion-binding site?

At pH 4, the cavity was closed off to a hydrated Na⁺ ion. In the current structure at pH 6.5, however, the cavity has opened up to a hydrated Na⁺ ion and residues in the pH sensing domain adopt a different conformation. We conclude that this is a pre-step in the alternating-access mechanism and necessary to achieve transport.

As far as we are aware, the lowest pH to detect Na⁺ binding is at pH 5.0 by SSM-electrophysiology when a pH gradient is applied (JBC 286: 23570). Essentially, this means that at pH 5.0 the cavity must be able to open up enough to enable a hydrated Na⁺ to bind to NhaA and be translocated, but it is not until a higher pH 6.5 that NhaA is re-protonated sufficiently to (efficiently) catalyze H⁺-efflux.

To solidify our main arguments, we have also made the following changes:

1. We show that at pH 6.5 one can detect significant Li⁺-catalyzed efflux at higher concentrations of substrate (150 mM as compared to 10 mM LiCl).
2. We identified new interactions in the pH sensor with residues His253, His256 and Lys153. Surprisingly, the mutation of residues to alanine had not been analyzed by the Padan lab. We now show that the mutation of His253Ala, His256Ala, and Lys153 to alanine or glutamine almost completely abolishes transport activity. Thus, the salt-bridges in the pH sensor are critical for NhaA activation.
3. Whereas wildtype NhaA reaches 50% activity at pH 7.5, the Glu78Ala mutation reaches 50% activity already pH 7.0 (acidic shifted by 0.5 pH units) The Asp11Ala mutation does not show 50% activity until pH 8.0 (alkaline shifted by 0.5 pH units). This data adds to the many other pH sensing mutants shown to alter the pH of activation. Notably, the calculated K_M for these mutants at pH 7.5 are similar to WT.
4. We now show water and Na⁺ accessibility data from MD simulations for the low and high pH structures that give a clearer graphical representation of ion accessibility.
5. We think the historical term “pH sensor” is a bit confusing as it was previously thought to explain the pH dependence of NhaA. Our new data is more consistent with this region acting as a “gate” and controlling the pH of the initial activity. Above the activation pH the transporter is rather operating on “Na⁺” vs “H⁺” competition, which is a property of the ion-binding site’s pKa. Taken together, we now refer to the region as a “pH gate”.

New Figure 2D.

Minor comments:

1) In the model in Fig.1, the cytoplasm is at the top while, in all the others, it is at the bottom. In Fig. 2, they used "Out and In" for cytoplasm and periplasm

Thank you. We have now corrected the labelling and orientation of NhaA so that the cytoplasm is always on the bottom.

2) From the Introduction: "... bacterial homologues are often" Can the bacterial antiporters be considered homologs of the human? They catalyze the same reaction, but they are very different functionally (electrogenicity and pH activation), and they share minimal sequence similarity, if at all.

Structurally the bacterial homologs (particularly those with 13-TMs) have shown to be very similar to the mammalian exchangers (The EMBO Journal (2020)39:4541-4559). Because the bacterial models are easier to manipulate, we have been able to test ideas and develop concepts regarding the determinants of ion-exchange. While NhaA was established as the model system for Na⁺/H⁺ exchange, from a structural perspective it is different in how it oligomerizes compared to the mammalian ones.

Nonetheless, NhaA was found to have a cardiolipin binding site between the protomers that Etana Padan showed was required for functional activity (Scientific Rep 27:17662). We have further seen an influence of negatively-charged lipids on homodimerization of mammalian NHEs too (The EMBO Journal (2020)39:4541-4559; NSMB 29: 108–120). Thus, while the details might differ, we think the bacterial exchangers are important test-beds for developing concepts. Here, we find it remarkable that residues located on the surface of NhaA can directly influence when NhaA becomes active by controlling accessibility to the ion-binding site. Is this relevant for the mammalian exchangers? I am not sure, but we do find a number of conserved histidine residues clustered on the cytoplasmic surface cavity of NHEs too (see new Figure added below). Also, it would not surprise me if such an intrinsic conformational switch regulated by pH is also utilized by other transporters.

3) Some editing and spell-checking are necessary.

Our apologies. We hope the new version is easier to follow with a better incorporation of the computational work in particular.

Reviewer #2 (Remarks to the Author):

The manuscript by Winkelmann et al. reports the X-ray crystal structure of the *E. coli* sodium proton antiporter NhaA at pH 6.5. The structure reveals novel details concerning the mechanism for pH activation in this system through structural comparisons with the inactive state of the same protein, determined at pH 4.0 (PDB: 4AU5). Accompanying detailed MD and biochemical assays further support the main novelty in the study that the main driver for pH activation is that protonation of residues in the pH sensor results in the opening of a channel that allows hydrated sodium to access the central ion binding site. This model supports previous work on this and related systems (all referenced in the study) that identified the histidines on TM9 as likely to be the physiological proton binding sites. Additionally, the study then explores the effect of pH on the mechanism of transport as it relates to the Asp164-Lys300 salt bridge. Here

the authors use their higher resolution structure at pH 6.5 to further detail the steps surrounding the breakage of this interaction following sodium binding to Asp163.

Overall, I found the present study has some interesting new information that does significantly add to the current literature on sodium proton antiporters, most notably the visualisation of the effect of pH activation. However, I also found the paper hard to read and somewhat disjointed between the two sections. At times I was also unsure what was novel and what was already known in the literature.

I very much liked the first section, which compares the structure of the activated vs inactive transporter (Fig. 1). However, the description lacked insight into the mechanism of activation. I understood that protonation results in the reorientation of the salt bridges from interhelical to intrahelical and the importance of the His256-Asp11 interaction. But I was left wondering whether the authors had pinpointed the protonated residues or worked out, via their MD analysis, the actual steps in the activation process. I assume that protonation of either His253 or His256 is the first event in the activation process, but I couldn't determine whether the MD showed this? Without this, it appeared to me that the current structure essentially supports the previous literature, which had already identified roughly where the pH sensor was located but hadn't to date recorded the steps involved.

Thank you for your constructive feedback. We have re-written the paper and better incorporated the MD simulations that I hope makes the paper easier to follow. With more than 10 years of extensive research on the pH sensor there is a lot of information that has detailed the location of the pH sensor, but its been unclear how its coupled to pH activation. In this case I think "seeing is believing" and it is an important structure to show how this rearrangement takes place in the active state.

Although Etana Padan's lab has amassed a large number of NhaA mutants the mutation of His253 and His256 residues to alanine had not been experimentally tested, although their shift in pKa values had been noted previously in MD simulations. We could not find any data on the mutagenesis of Lys153. Surprisingly, we find that the His256Ala and His256Ala mutations are almost completely inactive as well as the Ly153Ala and Lys153Gln mutants. Thus, the newly identified salt-bridges seem critical for function, probably as these residues are required to stabilize an open cavity. The other mutations tested in the pH sensor shifted the pH at reaching half-maximal activity, but the Km values were similar to wildtype. Taken together, the pH sensor is acting as a pH gate, which controls the activation pH, but once fully-open, activity can be modelled by simple H⁺ vs Na⁺ competition, i.e., the pKa of the ion-binding site.

I wondered whether the authors could use their all-atom MD simulations, detailed on page 8, to replicate the activation process, at least locally, and observe the structural rearrangements displayed in their structural comparison. Could a systematic protonation of the residues and resulting analysis identify the key sites on proton binding? These results may enable the authors to

pinpoint a primary protonation site for activation or whether the proton can bind to several places and still result in activation.

We have included new analysis of the MD simulations to highlight the accessibility of the structures modelled at pH 4 or pH 7.5 for Na⁺ vs water (as a proxy for H⁺). Notably, we start all simulations with the better resolved structure at pH 6.5, but change the external pH (see new figure added to referee 1 responses). Whilst we can obtain an overall picture and highlight the key residues required for activation (and experimentally test these residues, as described above), we cannot use MD simulations to dissect systematically the individual states: Using fixed charge MD (as we have done in this work) is infeasible because of the large number of titratable residues that we would need to simulate in different charge states. This leaves constant pH MD (CpHMD), i.e., MD where protonation states change dynamically, as was done in the 2016 Nat Comm paper [doi: 10.1038/ncomms12940], using the low pH NhaA structure as starting point. CpHMD of membrane proteins is still extremely demanding and even with current supercomputing resources, only 10–20 ns of simulated time is feasible. Although conformational sampling in these simulations is better than from a normal 20-ns MD simulation (because they are performed with pH replica exchange), they are in our opinion nevertheless not suitable to directly infer the order of protonation events (and we were not able to comment on this question in the 2016 paper either). The observation of the decrease in binding site accessibility at low pH in the 2016 CpHMD work is consistent with the new evidence shown here, but the current work clearly identifies the structural elements (namely, salt bridge “switches”) that drive a relatively subtle conformational change (which we could not identify in the earlier work).

The authors compare their new structure at pH 6.5 to their previous structure at pH 3.8. However, the structure at pH 3.8 was determined at 3.7 angstroms. Given the relatively minor changes in side-chain positions being discussed, how confident are the authors of the side chain locations in their previous structure?

Good point, we should have shown this. The electron density for our NhaA WT dimer was 3.7 Å, but the structure of the triple stabilization mutant (with WT activity) was at 3.5 Å. The electron density was good for this modest resolution as we had 4 molecules in the a.s.u and so we could apply NCS averaging (PDB ID: 4ATV). As shown in the new Supplementary Figure 1B, the electron density from the structure at pH 3.8 was of sufficient quality to model the residues of the key pH sensing residues.

Page 8 – the author's highlight that Glu252 changes side-chain conformations in line with pH activation. However, as drawn in Fig. 1C, the side chain does not appear to move very much at all?

Thank you for pointing out this inconsistency. The biochemical data that shows that the Glu252Cys mutant changes conformation with respect to pH was based on cysteine accessibility (Biol Chem. 2004 Jan 30;279(5):3265-72). Since a cysteine mutant would not be able to form electrostatic interactions with either His253 or Arg81, this observation was meant to highlight the region becoming more open at active pH. We have modified the text accordingly.

Page 9 – details the second part of the study, which focuses on the Asp164-Lys300 salt bridge. The authors focused on this interaction pertaining to the mechanism of sodium and proton binding in the central ion binding site.

I was very confused by this section, and I didn't understand what the authors had discovered. How does the temperature-dependent rate analysis link with the pH activation of the proton sensor? This section appeared to be included to explain the differences observed in previous papers of the effect of mutating Lys300 to Arginine, which differs between *E. coli* Nha and *T. thermophilus* NapA (Lys305), but why? It was also unclear how this explained the observation that a membrane potential could no longer drive transport via the Lys305Gln mutant. Is this linked to pH activation or the pH? The last paragraph of this section also seemed separate from the previous paragraphs. Again, I was unsure how measuring the melting temperatures resulted in further insight into pH activation, which is the paper's title?

We thank you for your comments. The Na⁺-sensitive salt-bridge and the argument that Lys300/Lys305 can act as a proton carrier has been a debated topic with a number of arguments and counter-arguments spanning a number of different publications as its an important fundamental question for the ion-exchange mechanism:

- G. Masrati, M. Dwivedi, A. Rimón, Y. Gluck-Margolin, A. Kessel, H. Ashkenazy, I. Mayrose, E. Padan, and N. Ben-Tal, "Broad phylogenetic analysis of cation/proton antiporters reveals transport determinants," **Nature Communications**, vol. 9, no. 1, p. 4205, 2018.
- O. Călinescu, M. Dwivedi, M. Patiño-Ruiz, E. Padan, and K. Fendler, "Lysine300 is essential for stability but not for electrogenic transport of the *e. coli* NhaA Na⁺/H⁺ antiporter," **Journal of Biological Chemistry**, vol. 292, pp. 7932–7941, 2017.
- M. Patiño-Ruiz, M. Dwivedi, O. Călinescu, M. Karabel, E. Padan, and K. Fendler, "Replacement of Lys-300 with a glutamine in the NhaA Na⁺/H⁺ antiporter of *Escherichia coli* yields a functional electrogenic transporter," **Journal of Biological Chemistry**, vol. 294, no. 1, pp. 246–256, 2019.
- Y. Huang, W. Chen, D. L. Dotson, O. Beckstein, and J. Shen, "Mechanism of pH-dependent activation of the sodium-proton antiporter NhaA," **Nature Communications**, vol. 7, p. 12940, 10 2016.
- J. A. Henderson, Y. Huang, O. Beckstein, and J. Shen, "Alternative proton binding site and long-distance coupling in *E. coli* sodium-proton antiporter NhaA," **Proc Natl Acad Sci U S A**, vol. 117, no. 41, pp. 25517–25522, 2020.
- P. Uzdavinyis, M. Coincion, E. Nji, M. Ndi, I. Winkelmann, C. von Ballmoos, and D. Drew, "Dissecting the proton transport pathway in electrogenic Na⁺/H⁺ antiporters," **Proceedings of the National Academy of Sciences**, vol. 114, no. 7, pp. E1101–E1110, 2017.
- C. Lee, S. Yashiro, D. L. Dotson, P. Uzdavinyis, S. Iwata, M. S. P. Sansom, C. von Ballmoos, O. Beckstein, D. Drew, and A. D. Cameron, "Crystal structure of the sodium-proton antiporter NhaA dimer and new mechanistic insights," **J Gen Physiol**, vol. 144, no. 6, pp. 529–544, 2014.

Nonetheless, it's a separate mechanistic question to the pH activation mechanism, but clearly an important question we could re-address given the significant improvement to the NhaA crystal structure from 3.5 to 2.2 Å resolution. In hindsight, however, we "glossed" over some of the background story that would not be so well known to those not working on Na⁺/H⁺ exchangers directly, and thus making this section harder to follow than needed. For example, it was counter-proposed that Lys305Gln showed electroneutral activity because it was unstable in detergent (JBC 294: 246, 2019.), rather than because Ly305 was a proton carrier (PNAS E1101, 2017). We had

therefore included melting curves of the Lys305Gln mutant to show that the stability of the protein was, in fact, similar to wildtype.

Based on your feedback, we have clarified that the Na⁺ sensitive salt-bridge is a separate question to pH sensing, and we focused on its breakage rather than if the lysine residue could also act as a proton carrier; as such we have removed the thermostability (melting) curves. We have further incorporated the biochemical data of salt-bridge mutagenesis with the MD simulations into a new Figure 4. We think the data nicely demonstrates that the salt-bridge can be a significant barrier for transport at room temperature, as shown here for NapA, which has evolved to operate at an optimum temperature of 65°C. That is, when the salt-bridge is removed by mutagenesis (Lys305Gln mutant), NapA now responds to temperature in a similar manner to NhaA that operates at 37°C. We couldn't test this theory directly by driving activity by a membrane potential as mutagenesis of the lysine residue to any other residue than histidine is electroneutral, i.e., so we had to drive activity by a ΔpH gradient instead (see below: PNAS 114:E1101).

[REDACTED]

We have had very positive feedback of the temperature dependent rates from biophysicists. These measurements are not trivial and took while for us to do these measurements so that liposomes were not leaky. We hope that the revision version better explains the data and its addition adds to the overall quality of the paper.

Finally, I was not convinced by the argument in the concluding paragraph that sodium proton antiporters have a unique mechanism for activation that involves regulating access to the central binding site. As described, the gate that allows accessibility would need to be separate from the transport domain, such that you could mechanistically separate the two functions. From reading this study, it seems to me that the pH sensor has to be protonated to release the salt bridge locks that enable the transporter to move. This seems more similar to the situation with the glutamate transporters, where the rate-

determining step is the unlocking of the interactions between the bundle and scaffold domains. Here the rate-determining step is the protonation of the pH sensor, which unlocks the transport for activity.

I see your point, but the main difference is that the low pH structure is not actually part of the transport cycle, but an off-cycle intermediate. To make this point clearer, we now refer to the pH sensor as a pH gate and we have made a new summary Figure 5. In other words, NhaA has an intrinsic pH gate that needs to open-up from low to high pH in order for cations to be able to access the binding site. Once it does, the exchange activity is then driven by the pKa of the binding site and, as such, this state is no longer part of the transport cycle.

Minor points:

Fig.1. The transporter is shown in the LacY orientation (cyto up) but in Fig. 2 (cyto down). I would correct this.

Thank you, this has now been corrected.

I would also check the consistency of labelling. As above, Fig. 1 (cyto/periplasm) in Fig. 2 (In/Out). Fig. 1C, TM5 is TMV. Fig 1D, helices need labelling and the sodium tunnel labelled.

Thank you. We have made the sodium tunnel clearer, but felt relabeling TMs was not needed for Fig. 1D as we didn't want to make the figure too busy.

Reviewer #3 (Remarks to the Author):

The manuscript by Winkelmann et al presents crystal structure of the Na/H antiporter NhaA at activating pH condition. Combining this structural model with molecular dynamics (MD) simulations, the authors address the mechanistic basis for pH sensitivity in this system. While the topic is very interesting, I find that presentation of the manuscript is not the most optimum, especially when it comes to description of the MD methods and results. Therefore, in the current form of the manuscript, I cannot properly evaluate the manuscript. Below I list several key aspects of the presentation that I think the authors should address to bring more clarity to their work:

1. I find it remarkable that only in the Methods section the authors describe so called "salt bridge mechanism" and "two Aspartate mechanism" and based on these mechanisms they present classification of states they simulate (all this in Methods only). This information is so buried in the manuscript that it is pretty much impossible to follow which states and mechanisms are probed in the simulations as they are presented in the main text. These mechanisms and the associated states (S1-S4) must be presented either in the Intro or in the Results,

and then when the MD data is described, each simulation condition must be associated with one of the states in the text. As it stands, the manuscript talks about high/low pH conditions in Results, S1-S4 states are defined in the Methods (and never used in the Results) and there is no connection between them (at least for a person who does not work on NhaA transporter).

You are right and we should have done a better job at incorporating the MD simulations. We have now added a lot more detail into the main text for the MD simulations and also rearranged the work to make it much clearer for the general reader and experts to follow. Following your suggestion, we also added a new Supplementary Table S3, which details the protonation states of the binding site residues and summarizes pH conditions together with the total charge of the funnel and the binding site. Throughout the text related to MD we make clearer, which state we are referring to.

2. The assignment of the protonation states is presented in an equally confusing way. I would recommend adding to Table S3 protonation states for each of the key residues. I do realize that the manuscript refers several times to the earlier publications on this topic (as well as for S1-S4 state definitions), but still, I believe that such fundamental mechanistic concepts should be re-emphasized in this manuscript as well. This will also help to determine the novelty of this work.

Good idea. We have now added a lot more detail into the main text for the MD simulations and also rearranged the work to make it much clearer for the general reader and experts to follow, e.g., as pasted below. Please see also our comment to (1) regarding the inclusion of a new Table S3 (instead of expanding the previous table of simulations, which is now Supplementary Table S4).

... To complement the analysis of static structures, MD simulations were carried out starting from the high-resolution structure determined at active pH (see Methods). In order to fully sample Na⁺ binding events and possibly capture conformational rearrangements due to pH we performed microsecond-long equilibrium MD with fixed protonation states of all titratable residues for a given pH (see Supplementary Table S3) instead of constant pH MD^{40,41}, which would correctly simulate fluctuating protonation states but which is currently limited to tens of nanoseconds of simulations. At active pH 7.5, the cytoplasmic funnel carried a total charge of -1 (Supplementary Table S3) and remained open so that both Na⁺ and water entered the binding site (Figure 2C, D), even if the ion-binding site was modelled with two protons bound (corresponding to a local pH 4 in the binding site, state "S1" in Supplementary Table S3) and cannot bind Na⁺. In contrast, under inactive pH conditions the cytosolic funnel contracted over the course of the simulation with concomitant changes in salt bridge interactions (Figure 2C) and increase of the total charge to +3, preventing Na⁺ access to the binding site (Figure 2D). Taken together, we conclude the "pH sensor" is working like a "pH gate", controlling access to the ion-binding site.

There are also several statements in the paper that requires clarification or revision:

1. The manuscript states (on page 7) that MD simulations display little conformational drift. But some RMSD traces in Supplemental figures 2 and 3 look not at all converged. So, I am not sure whether those simulations can be considered as stable/converged.

The RMSD traces somewhat exaggerate the impression of large changes over the course of the simulation: The reference structure is the crystal structure and so the initial point is not at zero (as is often shown in RMSD traces) and therefore the figure is already zoomed in on the range where fluctuations happen. Therefore, the increasing slope appears more pronounced as in "typical" RMSD figures for MD simulations. We opted to show all the detail instead of using the graph scale to obscure the behavior of the simulations. We agree that in Suppl Fig 3, the RMSD for the middle right run (S4_1, protomer B) may not have fully stabilized and in Suppl Fig 4 this may be the case for lowpH_1:B (middle right) and lowpH_2:A (lower left) but overall, all simulations are below 2.5 Å (including other ones not shown in the manuscript but included below for the review) and the majority of protomers looks stable as judged by RMSD.

Figure for review: **RMSD time series for all MD simulations that were not included in the Supplementary Information.** Each panel corresponds to a set of simulations, with protomer A in the left and protomer B in the right column. Each simulation ID (see Supplementary Table S4) is listed on the right hand side of each row.

2. On page 6, when discussing cardiolipin binding residues, the manuscript states that their data on R204/R205 mutants “is consistent with in vivo data that showed these residues were the most important for homodimerization and function”. To me, this is a pretty big leap. I do realize that there is an expectation that those Arg residues can bind cardiolipin and previously resolved dimer construct had a lipid-like density at the dimer interface which was attributed to

cardiolipin. But is there any data (especially in vivo) which shows that cardiolipin is important for dimerization and function?

We apologize this was not clearer. Etana Padan's lab showed that NhaA breaks down into monomers in a CDL-deletion strain and is unable to complement growth under salt-stress (Sci. Rep. 2019 Nov 27;9(1):17662). Etana further showed that the Arg205 and Arg206 mutations were also unable to complement salt-stress. Here we confirm that the arginine mutants also abolish thermostabilization with CDL.

3. On page 7, sentence "The loss of the TM2-TM9 interaction has likely caused a rearrangement of Lys249, His253 and His256 in TM9" is based on comparison of two frozen structures. It is hard to determine "causality" from such comparison. It is best to limit description to observed differences between the two structures.

You are right. We have removed this conjecture.

REVIEWERS' COMMENTS

Reviewer #1 (Remarks to the Author):

The revised version of the manuscript by Drew and collaborators satisfactorily addresses many of the issues I previously raised. The authors document a "pH gate". The transport model is described as a simple competition between Na^+ vs. H^+ , which is solely explained by the pK_a of the ion-binding site.

I think the authors may want to relate to the history of the pH sensor in NhaA. In this reviewer's view, the pH dependence shown for NhaA is not unique and was documented for many other *E. coli* antiporters (at least MdfA, AcrB, and EmrE). The dependence of the transport reaction on pH is very similar, with no activity at acidic pH and an increase to a maximum at the alkaline pH values and an approximate middle point at around 7.5–7.8. With various approaches, the pK_a of the carboxylic residues considered essential for coupling has been estimated to be at approximately 7.5, a value well within the range of the intracellular pH of *E. coli* cells. Thus, it seems that, regardless of their specific structures or mechanisms, the transporters have evolved so that they are exquisitely tuned to function at the very constant cytoplasmic pH maintained by *E. coli* cells. If the pK_a were too low, it would generate a protein that, at physiological pH, has already released the previously bound protons that binds substrate but that cannot couple the substrate flux to the proton gradient. If the pK_a were too high, substrate binding would be inhibited, allowing for very little activity at around the intracellular pH of *E. coli*.

Interestingly, such a pH dependence of the antiporters' activity may explain the reported involvement of these antiporters in bacteria and archaea in regulating the internal pH at alkaline pH values, as already suggested for MdfA and MdtM, two H^+ -coupled multidrug antiporters from *E. coli*. Thus, when the intracellular pH increases, so will the rates of the H^+ -coupled multidrug transporters and, as a result, increased rates of H^+ influx. The role of NhaA in pH regulation has been overstated. *E. coli* can grow well at alkaline pH without NhaA, and NhaB provided there is no sodium in the medium.

Table S2

I am not following all the mutants that modified the pH profile of NhaA, but my feeling is that were many more than those mentioned in TableS2. Please check. I thought the first one was a His in the cytoplasmic domain. Could it be His224 or 227 that is shown in the figure the authors sent?

Reviewer #2 (Remarks to the Author):

The authors have addressed my initial comments. The manuscript is now much easier to follow and the new data further support the insights gained into pH-dependent activation of Na/H antiporters. The proposed model for pH regulated allosteric modulation is novel for SLCs and it will be interesting to see how this idea develops in the field. I have no further comments.

Reviewer #3 (Remarks to the Author):

The authors satisfactorily addressed all my previous comments.

Crystal structure of the Na⁺/H⁺ antiporter NhaA at active pH reveals the mechanistic basis for pH sensing

Corresponding authors:
Oliver Beckstein
David Drew

We appreciate the positive response concerning our manuscript. We have carefully examined each remark and responded to all points below.

REVIEWERS' COMMENTS

Reviewer #1 (Remarks to the Author):

The revised version of the manuscript by Drew and collaborators satisfactorily addresses many of the issues I previously raised. The authors document a "pH gate". The transport model is described as a simple competition between Na⁺ vs. H⁺, which is solely explained by the pKa of the ion-binding site.

I think the authors may want to relate to the history of the pH sensor in NhaA. In this reviewer's view, the pH dependence shown for NhaA is not unique and was documented for many other *E. coli* antiporters (at least MdfA, AcrB, and EmrE). The dependence of the transport reaction on pH is very similar, with no activity at acidic pH and an increase to a maximum at the alkaline pH values and an approximate middle point at around 7.5–7.8. With various approaches, the pKa of the carboxylic residues considered essential for coupling has been estimated to be at approximately 7.5, a value well within the range of the intracellular pH of *E. coli* cells. Thus, it seems that, regardless of their specific structures or mechanisms, the transporters have evolved so that they are exquisitely tuned to function at the very constant cytoplasmic pH maintained by *E. coli* cells. If the pKa were too low, it would generate a protein that, at physiological pH, has already released the previously bound protons that binds substrate but that cannot couple the substrate flux to the proton gradient. If the pKa were too high, substrate binding would be inhibited, allowing for very little activity at around the intracellular pH of *E. coli*. Interestingly, such a pH dependence of the antiporters' activity may explain the reported involvement of these antiporters in bacteria and archaea in regulating the internal pH at alkaline pH values, as already suggested for MdfA and MdtM, two H⁺-coupled multidrug antiporters from *E. coli*. Thus, when the intracellular pH increases, so will the rates of the H⁺-coupled multidrug transporters and, as a result, increased rates of H⁺ influx. The role of NhaA in pH regulation has been overstated. *E. coli* can grow well at alkaline pH without NhaA, and NhaB provided there is no sodium in the medium.

Thank you for your feedback. We completely agree that the pKa of the ion-binding site explains the competition between Na⁺ vs H⁺ at active pH. However, Etana's pioneering studies on NhaA have shown from many different approaches that the cytoplasmic surface influences the pH of activation, which is why mutations on the pH sensor/gate can alter the pH for reaching maximal activity, even if the ion-binding site has not been touched. Essentially, what we can show here, is that the cytoplasmic pH gate controls at "what" pH the ion-binding site becomes accessible to ions. This is supported by the MD simulations. Even with an ion-binding site modelled at an active pH (aspartic acids deprotonated), there is no Na⁺ binding at low pH as the cytoplasmic funnel closes. Naturally, we are familiar with other proton-coupled efflux pumps, but as far as we are aware, closure of either the outward and inward-facing states as a function of pH has not been observed.

- We have modified the text in the second to last paragraph of the discussion to clarify this is what we meant in terms of a unique pH regulation mechanism for NhaA.

Like many organisms, *E. coli* must constantly work to maintain an intracellular pH, e.g., in order to colonize the human gut, it must be able to grow between pH 4.5 and pH 9 (J Bacteriol. 2007 Aug; 189(15): 5601–5607). We agree that there are other Na⁺/H⁺ antiporters in *E. coli* that have more of a house-keeping role than NhaA, which primary functional role is to remove Na⁺ under high salt-stress. This does not mean, however, that its not important to regulate NhaA activity by pH in the cytoplasmic side in addition to the ion-binding site. For example, under

oxidative damage, *E. coli* upregulate a K⁺/H⁺ exchanger KefC to acidify the cytoplasm to help deal to detoxify compounds (Biochemistry 2014, 53, 12, 1982–1992). Under these circumstances, H⁺ influx by NhaA would work against KefC if it was not inhibited by an acidic pH. It is further possible that this pH sensor region might further be influenced by cardiolipin binding to the NhaA homodimer, as the addition of cardiolipin affects the apparently affinity of ²²Na⁺ to NhaA (Scientific Reports volume 9, Article number: 17662 (2019)).

- We have clarified in the beginning of the discussion that NhaA has a role for pH regulation under environmental stresses and high salinity. We have further described in the discussion how cardiolipin binding may also work in concert with the pH gate to control activation.

Table S2

I am not following all the mutants that modified the pH profile of NhaA, but my feeling is that were many more than those mentioned in TableS2. Please check. I thought the first one was a His in the cytoplasmic domain. Could it be His224 or 227 that is shown in the figure the authors sent?

Yes, there more mutations, but not all were analyzed for their pH dependence. We have limited the Supplementary Table to the residues defined in the cytoplasmic pH sensor, which is the cytoplasmic half of TM 9, loop TM8-TM9 loop and the cytoplasmic end of TM2. The residue His225 is on the periplasmic side.

Reviewer #2 (Remarks to the Author):

The authors have addressed my initial comments. The manuscript is now much easier to follow and the new data further support the insights gained into pH-dependent activation of Na/H antiporters. The proposed model for pH regulated allosteric modulation is novel for SLCs and it will be interesting to see how this idea develops in the field. I have no further comments.

Thank you for input that improved the manuscript and we agree.

Reviewer #3 (Remarks to the Author):

The authors satisfactorily addressed all my previous comments.

Thank you for input that improved the manuscript.